# Active Reinforcing Fiber of Cementitious Materials Using Crimped NiTi SMA Fiber for Crack-Bridging and Pullout Resistance

**DOI:** 10.3390/ma13173845

**Published:** 2020-08-31

**Authors:** Eunsoo Choi, Ha Vinh Ho, Jong-Su Jeon

**Affiliations:** 1Department of Civil Engineering, Hongik University, Seoul 04066, Korea; hovinhha@mail.hongik.ac.kr; 2Department of Civil and Environmental Engineering, Hanyang University, Seoul 04763, Korea; jongsujeon@hanyang.ac.kr

**Keywords:** active reinforcing fiber, shape memory alloy, cold drawing, crimped fiber, NiTi SMA

## Abstract

This study investigated the recovery stress and bond resistance of cold drawn crimped SMA fiber using two different initial diameters of 1.0 and 0.7 mm. These characteristics are important to the active prestressing effect and crack-closing of the fiber. NiTi SMA fiber was used for the cold drawing, and then crimped shapes were manufactured with various wave heights. After that, tensile, recovery, and pullout tests were conducted. The cold drawn crimped fiber showed softening tensile behavior more clearly than the cold drawn straight fiber when not subjected to heating, whereas they had the same tensile behavior under heating. The recovery stress and the residual stress of the crimped fibers were less than those of the straight fiber with the same diameter. Moreover, crimped fibers with a large diameter and higher wave height would induce more recovery stress and residual stress. The maximum pullout resistance of the crimped fiber was a function of the wave depth, embedded length, yield strength, and flexural rigidity of the fiber.

## 1. Introduction

Cementitious materials are good materials for civil construction with low cost and convenience for shaping. However, their low tensile strength and resulting cracking are critical drawbacks [1,2,3,4]. To overcome these problems, reinforcing fibers have been introduced as a solution, and some of them successfully raise the ductility and tensile strength of concrete/mortar and bridge cracks, thus delaying fracture [5,6,7,8,9]. The most commonly used reinforcing fibers are steel and synthetic fibers. 

Steel fibers are cheap and provide enough energy dissipation after cracking; they are commonly used with several shapes, such as end-hooked, twisted, and crimped [10,11]. These shapes help increase the mechanical bond resistance or end-anchoring resistance of steel fibers. End-hooked fibers have shown slip-hardening behavior in pullout tests with a high pullout load [12]. However, the anchoring effect is concentrated at the ends regardless of length of the fiber. Therefore, some modified end-hooked fibers with more hooked bends have been studied [13,14,15]. The resistance of twisted fiber is generated by the untwisting torsional moment resistance in the whole inserted length of the fiber [16,17], while the resistance of crimped fiber is caused by multiple anchorages of repeating indentations [18]. Steel fibers provide resistance with plastic deformation and mechanical bonding with deformed shapes. 

Synthetic fibers are produced from petroleum, metal, carbon, or glass in a chemical process. The most common synthetic fibers used for cementitious materials are polypropylene, polyvinyl alcohol, glass, and carbon. Polypropylene and polyvinyl alcohol are cheap and easy to mix, while they have a low modulus and provide only chemical bonds. Glass and carbon have high tensile strength and elastic modulus, but their high cost is their main drawback [19]. Glass and carbon fibers cannot improve resistance by changing their shape as steel fibers can. However, they are small in diameter, so they provide relatively large surface areas for the distribution of frictional resistance [20]. Some of the synthetic fibers such as polypropylene and polyvinyl alcohol are inert in the high pH environment of the cementitious matrix while carbon fibers have a high melting point; thus, synthetic fibers can minimize the harm of fire or environmental effect [19].

These reinforcing fibers provide only passive resistance to pulling out or cracking because they are activated just after developing deformation. This mechanism is effective for crack-bridging or energy dissipation. However, it cannot fully prevent the initiation of cracking in cementitious materials because the fibers provide relatively low resistance to delay crack-initiation; they provide only chemical and frictional bond resistance, which is too low to delay cracking. 

Recently, reinforcing fibers made of shape memory alloys (SMAs) have been studied as active reinforcing fibers because they can provide prestressing before cracking [21,22,23,24,25,26], so they can delay crack-initiation. They also provide crack-closing capacity using the recovery stress because of the shape memory effect [27,28,29]. Therefore, the prestressing and crack-closing capacity can be classified as active effects that cannot be provided by passive fibers, namely steel and synthetic fibers. Therefore, such SMA fibers can be considered active reinforcing fibers. To guarantee that these active effects occur, the bond resistance of SMA fibers needs to be as strong as possible. Straight SMA fiber is seldom used because it only has frictional resistance, which is too weak in comparison with anchorage resistance [30,31,32]. Previous studies have suggested various shapes of SMA fibers, such as L-, N-, dog-bone, and paddle-shaped ones [33,34,35,36]. Of these types, the paddled SMA fiber is the most effective to provide end-anchoring action, and thus it is the most appropriate for crack-bridging [10]. However, it is not suitable for mass production because its end parts must be clamped one by one. Dog-bone-shaped SMA fibers are easily produced because they can be made by heating the end parts; the heating can be applied by exposure to fire or boiling water. However, unfortunately, dog-bone-shaped fibers induce relatively weak bond resistance because the geometric discontinuity at the end part is too small to provide high bond resistance in comparison with the paddled SMA fibers. 

Thus, a new shape of SMA fiber is required to overcome the drawbacks of the previously suggested SMA fibers. It should provide: (1) active effect of prestressing and crack-closing capacity; (2) enough bond resistance for crack-bridging; and (3) availability for mass production. This study suggests that crimped SMA fibers made from cold drawn NiTi SMA wires can fulfill these three requirements. The goal of this study was to investigate whether crimped SMA fibers have a high pullout resistance as well as mass producibility [37]. For this purpose, SMA wires of two different diameters of SMA wires were used to make the crimped fiber. Then, the tensile behavior and recovery stress of the crimped SMA fibers were investigated, and the bond behavior was assessed through pullout tests. 

## 2. Characteristics of Crimped SMA Fibers

### 2.1. Properties of NiTi SMA Fibers

Ni50.4-Ti (wt.%) SMA fibers with two diameters of 0.7 and 1.0 mm were used for cold drawing and the subsequent fabrication of crimped fibers. First, they were annealed to make them straight and remove dislocations, and then were cold down to room temperature. The cold drawing process started by heating the fibers at the temperature of 500 °C, which made them bulge and the diameters increased to 1.020 and 0.702 mm. After that, the fibers were cold drawn under room temperature of about 25 °C, and their final diameters became 0.955 and 0.660 mm, respectively. The reduction of diameters is summarized in Table 1, in which ARR represents area reduction ratio of each fiber. AR1.0 and AR0.7 refer to the two types of cold drawn SMA fibers with diameters of 1.0 and 0.7 mm, while N-AR1.0 and N-AR0.7 refer to the non-cold drawn SMA fibers, which are prior to the cold drawing, with the two diameters mentioned above. This work introduces prestrain along the fiber direction that can produce recovery stress in the fiber due to the shape memory effect by heating.

Cold drawing not only changed the dimensions of the fibers but also affected the temperatures of phase transformation. Figure 1 shows two differential scanning calorimetry (DSC) curves of the SMA before and after cold drawing. The DSC is a thermoanalytical technique, where the difference in the heating amount required to increase the temperature of a sample and reference is measured as a function of temperature. 

After cold drawing, the starting austenite state temperature (A_s_) and the finished austenite temperature (A_f_) of the AR1.0 fiber changed from −13.57 and 7.44 °C to 40.87 and 71.61 °C, respectively. For the AR0.7 wire, the temperatures of phase transformation increased from −11.7 to 42.01 °C (A_s_) and from 4.3 to 91.46 °C (A_f_). Thus, the transition temperature apparently occurs with more than 10% ARR; the phenomenon is similar to that observed in a previous study [38]. The cold drawn SMA wires with the starting temperature of austenite (A_s_) of 40.87 or 42.01 °C, which was higher than room temperature, can be stored safely without any deformation recovery at room temperature.

### 2.2. Production of Crimped Fibers

Crimped fibers were produced by using a rolling device, which is shown in Figure 2a. This device had a pair of crimper gears with 128 teeth; the top and bottom widths and the depth of each tooth were 0.2, 0.6, and 0.5 mm, respectively (see Figure 2b). The crimper gears with an outside diameter of 26 mm rotate about their centers and crimp SMA wires as they are fed through the rolling gears. By controlling the gap between the two gears, various wave heights of crimped fibers could be realized; a large gap created a shallow crimped fiber, and a small gap made a deep one (see Figure 3). The wave height was measured along wave direction, and the thickness was measured perpendicular to the wave. The difference between these two values is the wave depth. First, in this study, we made four types of crimped fibers with various wave heights by increasing 0.05 mm using the 1.0-mm AR wires. Table 2 shows their dimensions in thickness and wave-height. However, their wave heights were not controlled exactly because the gap between the two gears was manually adjusted. Thus, the values became 1.018, 1.039, 1.077, and 1.138 mm; they are referred to as CR1.0-1–CR1.0-4. The crimped fiber became thicker than the AR fiber due to the crimping because of Poisson’s effect; the thickness is measured at the thickest part perpendicular to wave direction. For example, the CR1.0-4 fiber was 0.017 mm thicker than the AR1.0 fiber. The increment in the thickness of the crimped fiber increased with more indentation; it seems that the crimping work presses the fiber and thus increases the thickness in the lateral direction. The wave depths of the crimped fibers ranged from 0.053 to 0.166 mm. The ratio of wave depth (WDR) to thickness of the crimped fiber is also presented in the table; the values varied from 5.5% to 17.4% for CR1.0-1 to CR1.0-4. Using the AR wire with 0.7-mm diameter, crimped fibers were also made following the WDRs of the CR1.0 fibers. The four WDRs of the CR0.7 fibers are also listed in Table 2. The values deviate from those of the CR1.0 fibers because the adjustment of the gap in the rolling device is too small to control exactly. For the CR0.7 fibers, the WDRs varied from 8.3% to 18.6% of the CR0.7-1 to CR0.7-4, and the corresponding wave depths varied from 0.055 to 0.124 mm.

The deformation recovery and stretching effect of the CR fiber strongly affected development of recovery stress, and geometrical variation of wave depth and thickness influenced bond resistance of the CR fiber. When the fibers are heated with flame without any restraint, an AR fiber shrinks because of deformation recovery of the shape memory effect. CR fibers also recover the prestrain induced by cold drawing, while the bent indentation induced by crimping is stretched and flattened. Thus, if the deformation recovery exceeds the stretching effect, the CR fiber will shrink. For the AR1.0 fiber, the length shrinking ratio (∆L/L) was 3.13%, as shown in Table 2, while the values of the CR1.0 fibers, which are smaller than that of the AR1.0 fiber, decreases with more indentation, which indicates that a greater wave depth leads to a greater stretching effect. The same trend in length shrinking was observed for the AR0.7 and CR0.7 fibers, while the CR0.7-1 and CR0.7-2 fibers showed similar values because the two fibers had very similar wave depths. When subjected to heating, the wave depth of indentation of the CR fiber decreased due to the stretching effect, as shown in Figure 4a, whereas the thickness of the CR fiber bulged because of Poisson’s effect (see Figure 4b). 

### 2.3. Tensile Behavior of the Fibers

Tensile tests were conducted on the AR fibers with two different initial diameters as well as on the CR fibers, namely, AR1.0, AR0.7, CR1.0-1 to CR1.0-4, and CR0.7-1 to CR0.7-4. The tests were controlled by displacement control with a loading speed of 1.0 mm/min. under room temperature of approximately of 25 °C. A fiber with a length of 50 mm was gripped by two cramps 5 mm from each end of fiber; the test set-up is presented in Figure 5.

The results are presented in Figure 6a,b, which shows the tensile behaviors of the fibers that were not subjected to heat treatment. Additionally, the AR and CR fibers were heated to induce phase transformation, and then tensile tests were conducted on them; the results are shown in Figure 6c,d. The stress of crimped fibers at the yield strain of AR fiber is shown in Table 3. The SMA fiber with a larger diameter showed hardening tensile behavior more clearly than that with a smaller diameter; these results do not depend on the shape or heat treatment.

Without heating treatment, the cold drawn fibers did not show phase transformation. The stress–strain curves of the AR1.0-N fiber, where “-N” indicates the case without heat treatment, showed the initial linear part which increased up to 1.0% with the secant modulus of 22.6 GPa, and the following part which showed softening behavior with a secant modulus of 12.0 GPa until yielding at the strain of 8.7%. After the yield, the AR1.0-N showed plastic behavior until failure. The AR1.0-N and CR1.0-N fibers showed almost the same tensile behavior up to 1.0% strain. Then, the crimped fibers more clearly showed lesser stiffness than AR fiber due to the stretching effect at the crimped part. At the strain of 8.7%, the AR fiber yielded at 950 MPa while the CR1.0-1N to CR1.0-4N fibers had stresses of 863, 840, 800, and 775 MPa, respectively. The crimped fiber with a larger wave height showed lower stress. The crimped fibers showed continuously increased stresses by 9–18% before yielding. The secant modulus of the 0.7-mm diameter fibers was 22.6 MPa at the strain of 1.0%, which is equal to that of the AR1.0-N fiber. However, after the initial elastic part, the AR0.7-N fiber had a Young’s modulus of 10.5 GPa, which is lower than the AR1.0-N fiber modulus. The AR0.7-N fiber yielded at 730 MPa of 6.3% strain; these values were also lower than that of the larger one. Therefore, in the plastic part, the smaller diameter AR fiber was softer than the larger AR fiber. The effect of crimping on the tensile behavior of the fibers with 0.7-mm diameter was similar to those of 1.0-mm diameter. The stresses at 6.3% strain of the CR0.7-N fibers were 6–18% lower than the yield stress of the AR0.7-N fiber.

With the application of heating, the fibers shrunk along the fiber direction. The strain at ultimate stress decreased by 10% with the CR1.0-H fibers and 5% with the CR0.7-H fibers, while the ultimate strain of the AR fibers did not change. This indicates that the decrement of ultimate strain was significant for the crimped fibers due to their shape. The influence of diameter on the phase transformation point of cold drawn fibers under heat treatment is seen clearly in the initial zone of the tensile curve. The 1.0-mm diameter heated fibers showed phase transformation early at about 1.5–3% strain with the stress of nearly 400 MPa, while the 0.7-mm diameter heated fibers showed phase transformation with 3–6% strain at 500 MPa. After that, they showed phase transformation with softening behavior. The 1.0-mm diameter fibers yielded at 966–1006 MPa, and the 0.7-mm diameter fibers yielded at 766–833 MPa. These yield values were higher than those for the fibers not subjected to heating.

### 2.4. Recovery Stress of the Crimped Fiber

#### 2.4.1. Test Set-Up

The recovery test of AR and CR fibers was set up as shown in Figure 7. A 50-mm-long fiber was gripped by two cramping devices at both ends, and then a chamber was placed on the fiber to cover it. A type k thermocouple, which is the blue electrical wire in Figure 7, was wound on the fiber surface to check the temperature variation. One side of the chamber was open to allow hot air to be blown into it from a hot gun; thus, the chamber contains the hot air. For each type, a fiber was heated to temperatures of 100, 150, 200, or 300 °C, and then it was cooled to room temperature, which was about 25 °C. The starting temperature of 100 °C was chosen because it was a little higher than the A_f_ of the SMA fibers, and temperature increment was 100–300 °C, which was enough to complete phase transformation of the SMA fibers. Meanwhile, the tests with 150 °C were added because the specimens of pullout and compressive tests were heated at the temperature. Recovery stress was induced when the fiber was heated, and the residual stress remained when the fiber was cooled down to room temperature. 

#### 2.4.2. Recovery and Residual Stress

The temperature–stress curves of the fibers are provided in Figure 8. Generally, the recovery stress and the residual stress at 100 °C were significantly lower than those at the other higher temperatures. It is deemed that the fibers did not undergo complete phase transformation because the heating was stopped just after the type k thermocouple reached 100 °C, and thus all parts of the fiber seemed not to arrive at 100 °C. Therefore, it seems that all parts of the fiber may not have undergone phase transformation. This means that a heating temperature higher than 100 °C can induce more phase transformation and thus increase the recovery stress. Meanwhile, after the phase transformation of the fiber is completed, the recovery stress does not increase any more, and more heating beyond this point would decrease the measured stress because of thermal expansion of the fiber. For the AR1.0 fiber, the stress reached the maximum at around 230 °C (see Figure 8a). After that, the stress decreased with increasing temperature because of thermal expansion of the fiber. The same phenomenon was observed in the AR0.7 fiber, whose maximum stress was attained at around 160 °C (see Figure 8b). Thus, the maximum recovery stress of the AR fiber became almost stable after a specific temperature was reached, and the maximum recovery stresses were 388 and 270 MPa for the AR1.0 and AR0.7 fibers. Considering the cross-sectional areas of the AR fibers, the AR1.0 fiber produced approximately 2.9 times more tensile force with the maximum recovery stress of that of the AR0.7 fiber. Meanwhile, the residual stresses of both AR fibers increased with increasing heating temperature. It is considered that the fibers almost accomplished full phase transformation at temperatures over 150 °C. The positive effects of temperature at 200 and 300 °C were similar with slightly increased maximum recovery stress and residual stress in comparison with those at 150 °C. Higher heating temperature results in larger residual stress, which can be used for prestressing and crack-closing. However, high temperature also may deteriorate the characteristic of mortar or concrete [39]; thus, heating to around 150 °C is appropriate for SMA fiber reinforced specimens, although, for the prestressing effect, the fiber could be heated with a higher temperature to have more benefits. 

The crimped fibers produced low recovery stresses and residual stresses compared with the AR fibers with the same initial diameter because crimped fibers are stretched by heating, which reduces the recovery and residual stress of the CR fiber. Thus, a larger wave depth induces a great reduction of the recovery and residual stress. As shown in Figure 9, the CR1.0 fiber with the largest wave depth (CR1.0-4 fiber) showed the lowest recovery and residual stress. The maximum recovery stresses and residual stresses of CR1.0-1 to CR1.0-4 were about 60–85% and 23–47%, respectively, in comparison with those of AR-1.0. For the CR0.7 fibers, the trend of the recovery and residual stresses with increasing temperature differed. The CR0.7-1, CR0.7-3, and CR0.7-4 fibers showed similar recovery and residual stresses, while the CR0.7-2 fiber showed the highest values; this is totally different from the trend of the CR1.0 fibers.

For both AR and CR fibers with 1.0-mm diameter, the recovery stress as well as the residual stress significantly depended on the temperature. The increasing trend of temperature took the increment of recovery stresses and residual stresses. Moreover, the crimped fiber with a lower wave height (straighter) showed higher values. All crimped fibers had lower stresses than the AR fiber because of the stretching effect. The maximum recovery and residual stresses of the CR0.7 fibers also showed an increasing trend with increasing temperature. However, the maximum recovery and residual stresses of the CR0.7 fibers are not proportional to the wave height as in the CR1.0 fibers. It seems that manufacturing of the CR0.7 fiber was not conducted exactly to control the wave height because the gap between the two gears is too small to precisely control. Thus, it is may possible for the CR0.7-2 fiber to show the largest recovery and residual stress among the CR0.7 fibers. The detailed values of the fibers are shown in Table 4, and the maximum values of recovery stress and residual stress are compared in Figure 9. In the tests, it should be notified that the temperature of type k thermocouple is exactly same as that of the heated fiber because the heating was stopped when the temperature of the type k thermocouple reached the target value; thus, the temperature of the fiber may be lower than that of the type k thermocouple. For the heating cases with 100 °C, the fibers may be not experienced complete phase transformation. If a whole fiber undergoes the phase transformation, the recovery stress becomes stable and decreases with more increasing temperature because of thermal expansion of the fiber; the decrement in the recovery stress is observed for the heating cases of 200 °C and 300 °C.

## 3. Pullout Test and Results

### 3.1. Specimens and Test Set-Up

Each SMA fiber was made by cutting a long SMA wire to length of 30 mm, and half of the length was embedded in the mortar matrix in a specimen with half dog-bone shape. The properties of crimped fiber are presented above through the experimental tests. The compressive strength of the mortar is 55 MPa, and composition of the mortar is shown in Table 5.

For each type of fiber, six specimens were prepared, and half of them were heated in an oven at 150 °C to induce the shape memory effect of the fiber. A type k thermocouple (blue electrical wire in Figure 10a) was wound on the surface of the fiber embedded in the mortar matrix to measure the temperature. Figure 10b shows time–temperature history of the heated specimen. The temperature of the fiber buried in mortar increased rapidly for 1 h. After that, the increment of temperature became blunt for 2 h and stable with 145 ℃ after 3 h. The duration with the temperature beyond 91.46 °C (A_f_) ensured that the phase transformation of the fibers was perfectly completed.

The dog-bone-shaped part of the specimen was gripped at the bottom, and the fiber was hold by the wedge grips 5 mm from the top surface of the specimen. Figure 11 presents the test set-up and dimensions of a specimen as well as a photograph of a specimen after the test was completed. The pullout force was measured by a load cell at the top of the machine, and the displacement of the fiber was measured by the stroke of the actuator. The pullout speed was set at 1.0 mm/min, and the sampling rate of response was 5.0 Hz.

### 3.2. Pullout Behavior

The force–displacement curves of the specimens and their averages are presented in Figure 12 and Figure 13. The stress value was calculated by dividing the measured pullout force by the cross-sectional area of the AR fiber. The symbols “N” or “H” following the name of fiber denote non-heated and heated specimens, respectively. For example, CR0.7-1H indicates that first specimen with cold drawn fiber with a 0.7-mm diameter underwent heat treatment before the pullout test. 

#### 3.2.1. General

AR fibers are straight; thus, there is only chemical and frictional resistance. At peak bond resistance, the chemical resistance is broken, and then the frictional resistance decreases with pulling out because the total frictional force is proportional to the embedded length. The AR0.7 fiber showed the typical behavior of a straight fiber. However, the AR1.0 fiber showed plateau force after the peak; it seems that the sharp end cut of the fiber provides end anchoring resistance that is constant regardless of the embedded length. When AR fibers are heated, they bulge due to the shape memory effect, and the bulging induces confining pressure around them, resulting in increased frictional resistance. Thus, the heated AR fibers showed higher bond resistance than non-heated AR fibers. The CR fibers showed wave-like behavior in their force–slip curves because of the indentation of the CR fiber. Thus, the wavelength of the pullout behavior is very close to the wavelength of the CR fiber. Peak forces and the corresponding displacements are presented in Table 6.

#### 3.2.2. Wave Depth Effect on Pullout Resistance

The relation between maximum pullout resistance and wave depth of crimped fibers was showed in Figure 14. For the CR1.0-N fibers without heat treatment, a larger wave depth generally produced a greater maximum pullout force, which occurred at the first peak, because deeper indentation can produce more mechanical resistance. The maximum pullout force is not linearly proportional to the wave depth of fiber. The CR1.0-1N and -2N fibers did not experience yielding, while the CR1.0-3N and -4N fibers yielded. In particular, it seems that the maximum pullout force of the CR1.0-3N fiber just exceeded the yield stress of the fiber because the hardening range after yielding is relatively short. Thus, the critical wave depth of the CR1.0 fiber for yielding would be just below 0.105 mm. After yielding, the CR1.0 fiber showed hardening behavior; thus, the CR1.0-4N showed slightly higher maximum pullout resistance than the CR1.0-3N fiber. 

The wave depth effect of the CR0.7-N fiber was similar to that of the CR1.0-N fiber; a deeper wave produced higher pullout resistance. However, not all of the CR0.7-N fibers showed yield behavior. Moreover, even the CR0.7-4N with the largest wave depth did not show apparent yield behavior; the apparent yielding of a fiber disturbs (for example, CR1.0-3N) or eliminates (for example, CR1.0-4N) the wave pattern in pullout behavior. The #1 and #2 fibers of the CR0.7-4N just exceeded the yield stress of the fiber. Thus, it is inferred that the critical wave depth of the CR0.7-N fiber for yielding was around 0.124 mm. 

A comparison of the two types of fibers showed that the CR0.7-N fiber does not produce larger pullout force with larger wave depth. For example, the wave depth difference between the CR1.0-1N and -3N was 0.052 mm, which resulted in increasing 261.5 N of pullout force, while the wave depth difference between CR0.7-1N and -4N was 0.069 mm, and increment of pullout force was just 38.7 N. Thus, increment of pullout force per unit wave depth of the CR1.0-N fiber was almost 10 times that of the CR0.7-N fiber. This phenomenon is highly related to flexural stiffness of the bent part in a crimped fiber. The flexural stiffness of the bent part is a function of wave depth, which acts as a cantilever arm in bending, and flexural rigidity (EI) of the fiber. Thus, if both types of CR0.7-N and CR1.0-N fibers have the same wave depth, the flexural stiffness of the CR1.0-N fiber is four times larger than that of the CR0.7-N fiber. Therefore, it can be said for the crimped SMA fiber that the maximum pullout force is highly related to the flexural stiffness of the bent part. 

#### 3.2.3. Pullout Resistance of Each Wave of Indentation

The peak pullout resistance of each peak as a function of displacement is shown in Figure 15. For non-heated specimens of the CR1.0-N and CR0.7-N fibers, the peak value almost linearly decreased with increasing pullout displacement excluding the last value, which is from the end part of a fiber and thus unstable. The difference between the adjacent two peak values indicates the pullout resistance that is provided by each wave of indentation. The differences are listed in the last three columns of Table 6; ∆_1-2_ indicates the difference between the first and second peak, ∆_2-3_ denotes the difference between the second and third peak; and ∆_avg_ is the average of the two differences. 

The average value ∆_avg_ represents the pullout resistance of each wave in a fiber. Thus, for the CR1.0-1N fiber, if the embedded length is increased by two more waves with 21.6 mm length, the maximum pullout resistance would be 725.9 N, which is close to the yield force. For the CR1.0-3N fiber, if one wave is eliminated from embedment, the maximum pullout force would diminish to 638.5 N, which is below the yield force of the fiber. The estimated value of 638.5 N is very close to the second peak value of 663.6 N; thus, this indicates that the maximum pullout force of a crimped SMA fiber can be controlled by the fiber-length, i.e. the number of waves. The previously studied dog-bone-shaped and paddled SMA fibers do not significantly increase the pullout resistance with increasing the fiber-length because they mainly provide pullout resistance by end-anchoring action. 

Figure 16 plots the increment of pullout resistance as a function of wave depth. In Figure 16a, for the CR1.0-N fiber, the result of the CR1.0-4N is not included because the fiber showed apparent yield behavior, and the result of the CR0.7-1N fiber was excluded because of its deviation. The relationship between the two values is almost perfectly linear, which means that the increment of pullout resistance is linearly proportional to wave depth until the yield of the fiber. Thus, SMA wire with even a diameter of 0.7 mm can be used to make crimped SMA fibers; if an SMA wire is too thin, it is difficult to make indentions on the wire. In this study, we tried to make crimped fibers using SMA wire with a 0.5-mm diameter, but this attempt failed because such small indentations could not be made on such small SMA wire. As shown in Figure 16, a fiber with larger wave depth under the same wavelength induces more resistance from the mortar to be pulled out. Another noticeable observation is that the slopes of the relationship for the two fibers are significantly different. The slope of the CR1.0-N fibers is 649.42, which is 2.45 times larger than the slope of the CR0.7-N fibers (265.26). As mentioned above, this is also related to the flexural stiffness of the bent part of a fiber. The range of wave depth for the CR0.7-N fibers was from 0.061 to 0.124 mm, which is a little larger than that of the CR1.0-N fibers, from 0.053 to 0.105 mm, while its increment of pullout resistance was less than half that of the CR1.0-N fibers. Thus, the increment of pullout resistance is also highly related to the flexural stiffness including flexural rigidity, EI. 

#### 3.2.4. Heating Effect on Pullout Resistance

When a crimped SMA fiber is heated, the fiber is stretched in the crimped part, and bulges in thickness in the lateral direction due to the shape memory effect. The stretching reduces the mechanical resistance during pulling out, while the bulging conversely increases frictional resistance. For the CR1.0 fibers, their pullout resistances decreased as a result of heat treatment; this indicates that reduction due to stretching exceeds increasing due to bulging. However, for the CR0.7 fibers, their pullout resistances increased as a result of heating because additional frictional resistance due to bulging overcomes reduction due to the stretching. The heated specimens of CR0.7-3H and -4H showed apparent yield behavior, while their corresponding non-heated specimens did not show apparent yield. It is conjectured that mechanical resistance in thick crimped fiber is dominant while frictional resistance is critical for the thin crimped fiber. The non-heated specimens showed unstable response with fluctuation because the summit of the indentation was pulled out discontinuously; thus, the response was up and down. However, for the heated specimens, the frictional resistance worked continuously along the fiber, and thus the response became smooth. 

## 4. Conclusions

In this study, we prepared four types of crimped fiber using cold drawn SMA wires with diameters of 0.7 and 1.0 mm with various wave heights of indentation. They were tested to measure their recovery and residual stresses and to observe their tensile behavior before and after heating. Finally, pullout tests were conducted to assess the bond resistance of the crimped SMA fibers considering the shape memory effect induced by heating. Based on the results of the experimental investigation presented, several conclusions can be drawn:CR fibers show more softening behavior in tension in comparison with AR fibers. However, after heating to induce phase transformation, the CR and AR fibers show almost the same tensile behavior.Crimped fibers by heating show lower recovery stress and residual stress in comparison with AR fiber having the same initial diameter because of the stretching effect. For CR1.0, the fibers with lower wave height showed higher values of recovery and residual stresses; however, the stresses of CR0.7 fibers were disturbed with increasing temperature.For both CR1.0-N and CR0.7-N, a larger wave depth induced a greater pullout resistance until they yielded. The critical wave depths of the CR1.0-N and CR0.7-N fibers for yielding were around 0.105 and 0.124 mm. If the pullout resistance exceeded the yield stress of a fiber, the wave pattern in pullout behavior was disturbed or eliminated.The maximum pullout resistance is a function of wave height (or wave depth) as well as the flexural rigidity of the fiber under the same fiber length; the wave depth and flexural rigidity consist of the flexural stiffness at the bent part in crimped fiber. The flexural rigidity (EI) of CR0.7-N was four times lower than that of CR1.0-N. Therefore, CR0.7-N was easily stretched during the pullout process and provide relatively lower pullout resistance.For the non-heated fibers, the peak pullout resistance decreased linearly with decreasing number of waves; this indicates that each wave produces constant pullout resistance. Moreover, the increment of pullout resistance due to increasing wave depth is also linear until yielding. Thus, for the CR1.0-N fiber, in the wave depth range from 0.053 to 0.105 mm, the maximum pullout resistance could be estimated for a fiber with specific wave depth and fiber length. The same assessment was possible for the CR0.7-N fiber with wave depth ranging from 0.061 to 0.124 mm.

Thus, the maximum pullout resistance can be a function of wave depth, embedded length, and yield stress of the fiber. The compressive strength of cement matrix can also be a factor that influences the pullout resistance of crimped SMA fibers. Based on the results, it can be concluded that the pullout behavior of a crimped SMA fiber is very complicated and affected by several factors. More detailed and intensive studies are required to clarify the influence of the factors mentioned in this work. 

## Figures and Tables

**Figure 1 materials-13-03845-f001:**
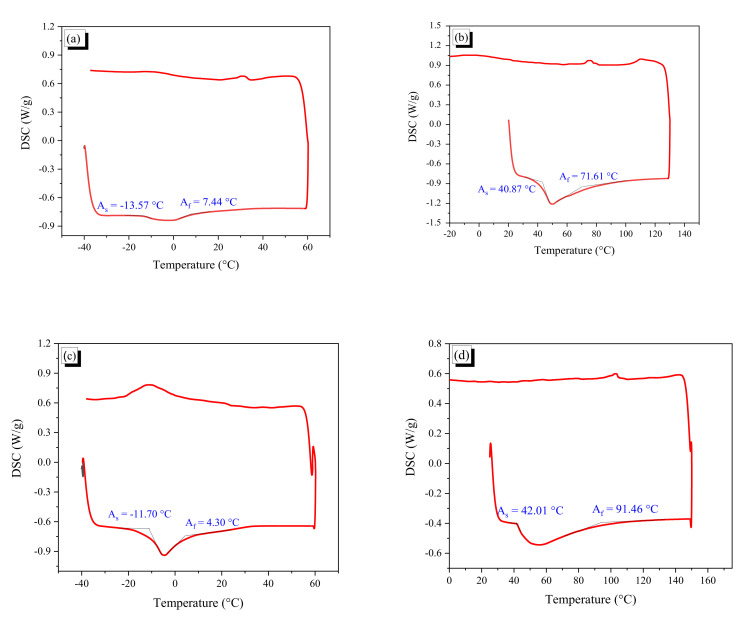
DSC curves of SMA fibers before and after cold drawing: (**a**) AR1.0 fiber before cold drawing; (**b**) AR1.0 fiber after cold drawing; (**c**) AR0.7 fiber before cold drawing; and (**d**) AR0.7 fiber after cold drawing.

**Figure 2 materials-13-03845-f002:**
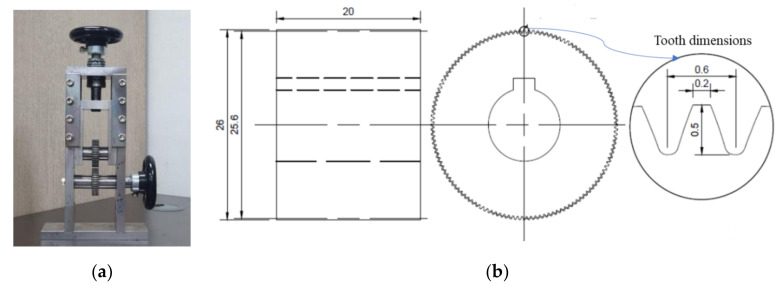
Rolling device with the dimension of the gear: (**a**) rolling device; and (**b**) dimension of the gear.

**Figure 3 materials-13-03845-f003:**
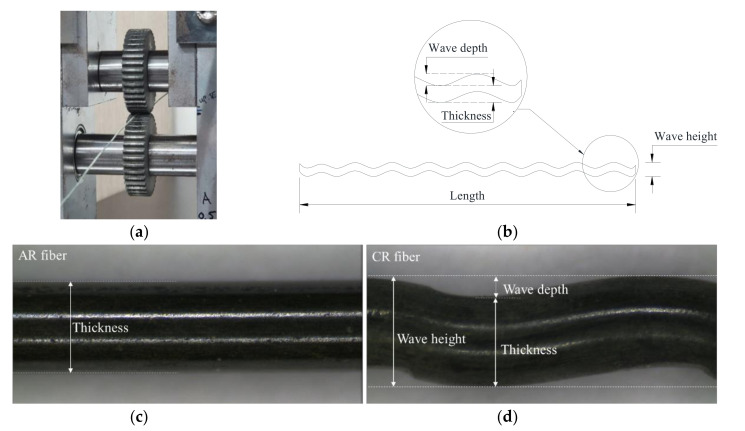
Crimping process and shape of crimped fiber: (**a**) crimping process; (**b**) shape of a crimped fiber; (**c**) photo of AR fiber; and (**d**) photo of CR fiber.

**Figure 4 materials-13-03845-f004:**
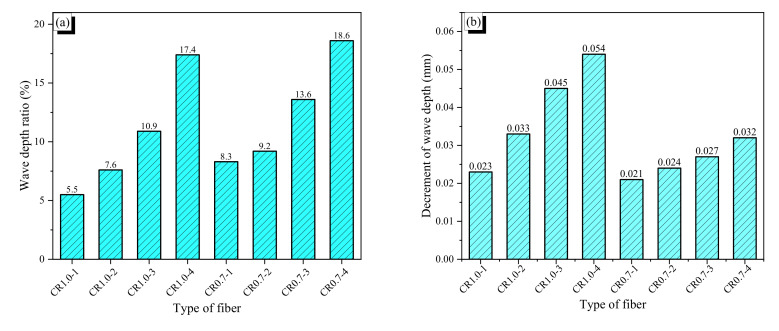
Dimension ratios of fibers: (**a**) wave depth ratio; (**b**) decrement of wave depth; (**c**) thickness increment ratio; and (**d**) length decrement ratio.

**Figure 5 materials-13-03845-f005:**
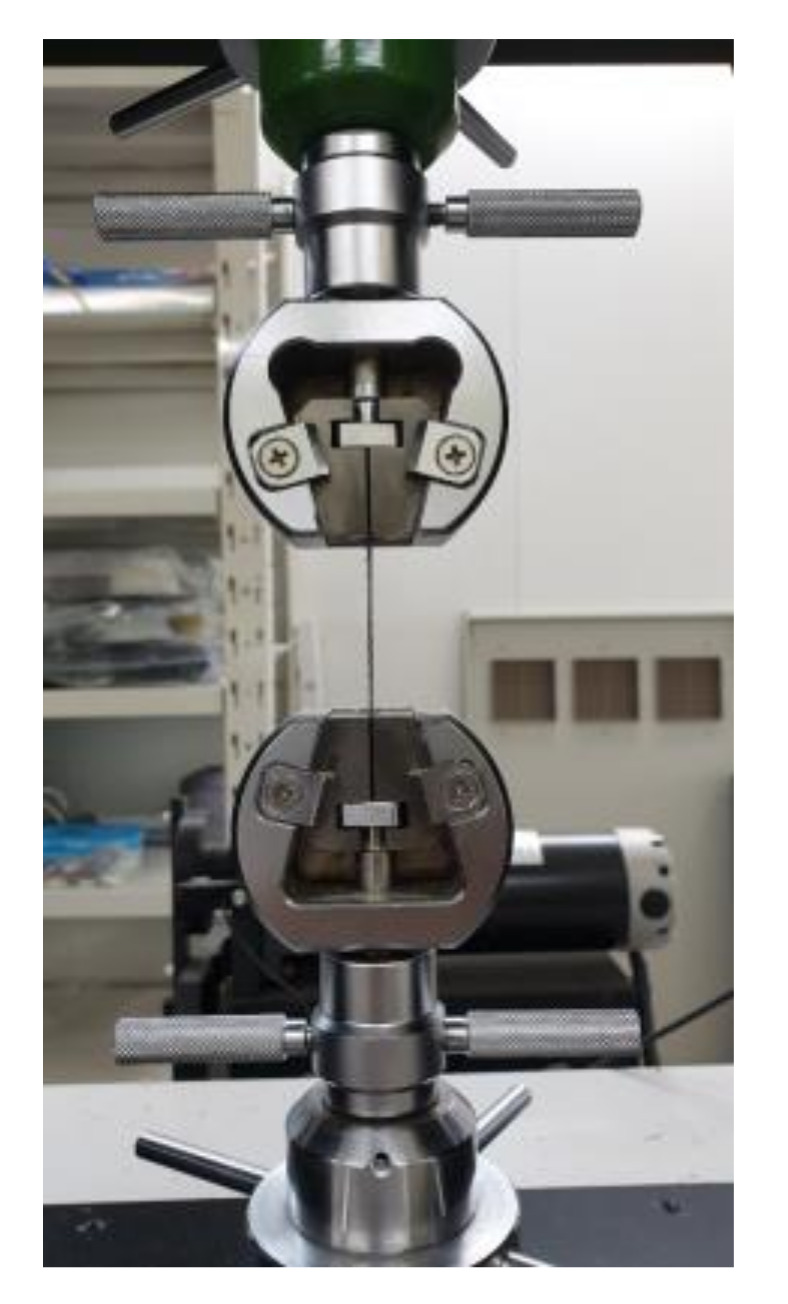
Monotonic tensile test setup.

**Figure 6 materials-13-03845-f006:**
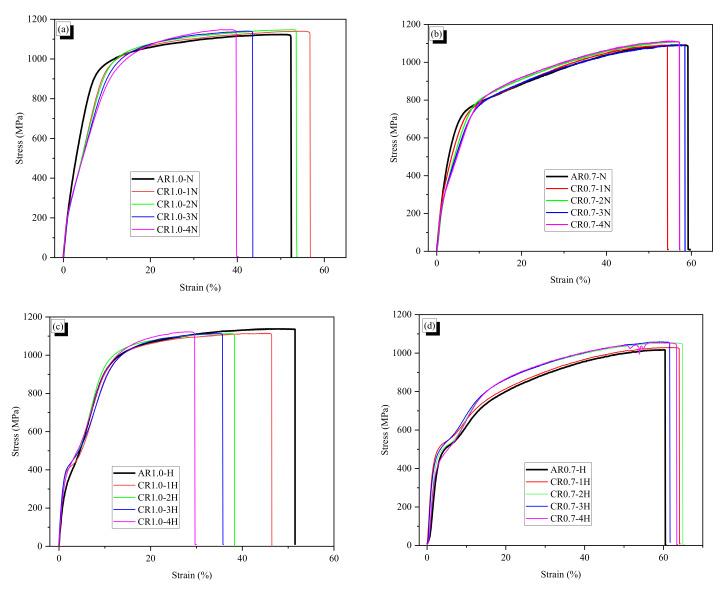
Tensile behavior of AR and crimped: (**a**) non-heated 1.0-mm diameter fibers; (**b**) non-heated 0.7-mm diameter fibers; (**c**) heated 1.0-mm diameter fibers; and (**d**) heated 0.7-mm diameter fibers.

**Figure 7 materials-13-03845-f007:**
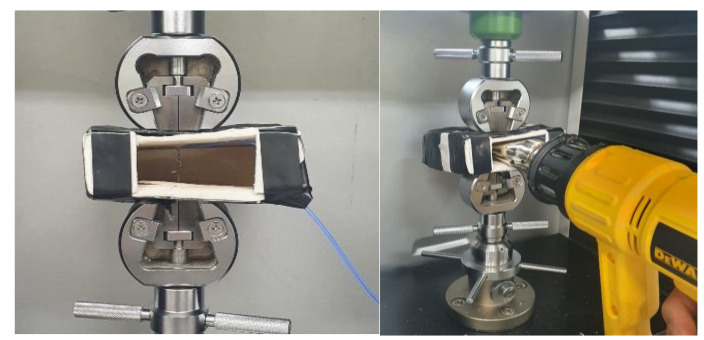
Test set-up to measure recovery stress of AR and CR fibers.

**Figure 8 materials-13-03845-f008:**
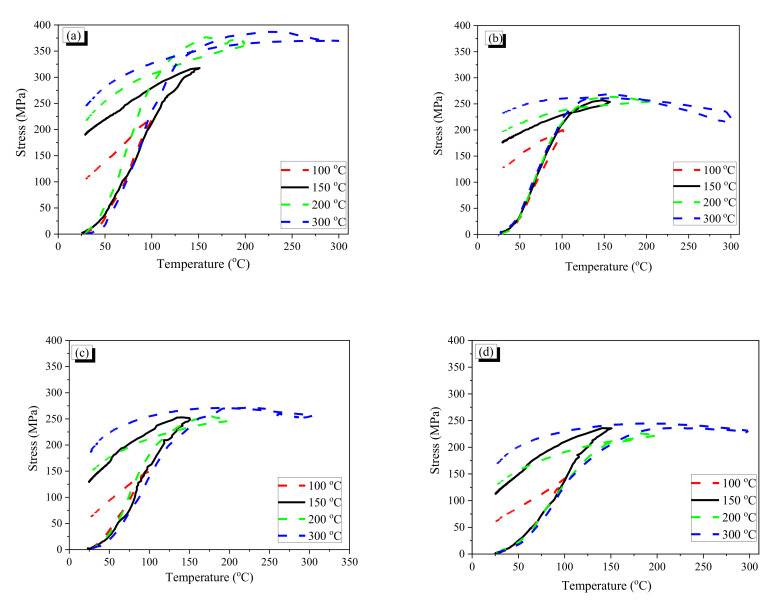
Temperature–stress curves of AR and crimped fibers: (**a**) AR1.0 fiber; (**b**) AR0.7 fiber; (**c**) CR1.0-1 fiber; (**d**) CR1.0-2 fiber; (**e**) CR1.0-3 fiber; (**f**) CR1.0-4 fiber; (**g**) CR0.7-1 fiber; (**h**) CR0.7-2 fiber; (**i**) CR0.7-3 fiber; and (**j**) CR0.7-4 fiber.

**Figure 9 materials-13-03845-f009:**
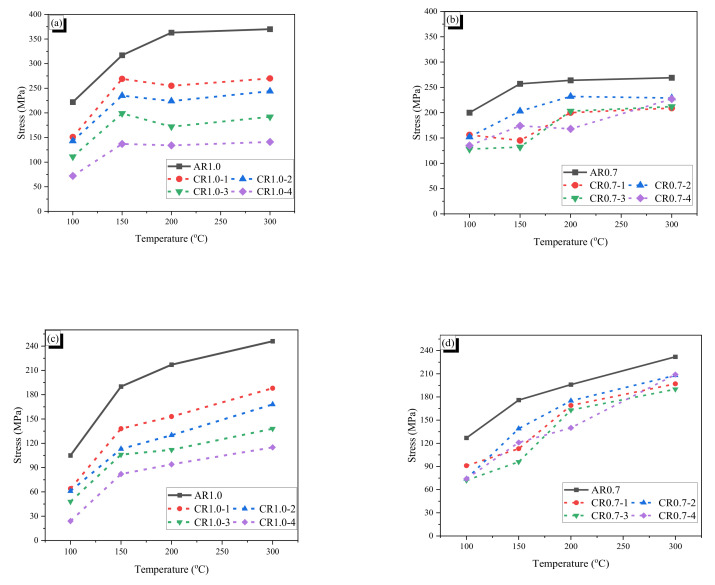
The comparison of the max recovery stress and the residual stress: (**a**) max. recovery stress of 1.0-mm diameter fibers; (**b**) max. recovery stress of 0.7-mm diameter fibers; (**c**) residual stress of 1.0-mm diameter fibers; and (**d**) residual stress of 0.7-mm diameter fibers.

**Figure 10 materials-13-03845-f010:**
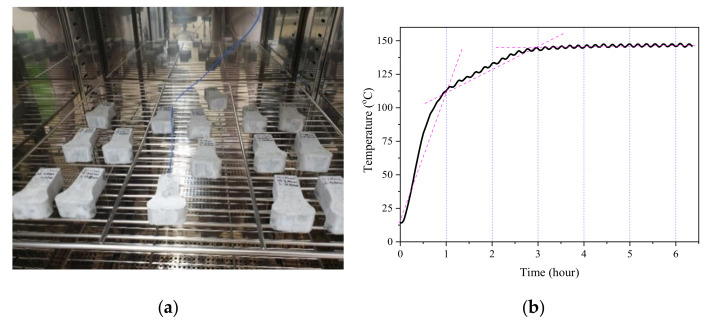
Time–temperature history of the heated specimens: (**a**) heating specimens; and (**b**) time–temperature history.

**Figure 11 materials-13-03845-f011:**
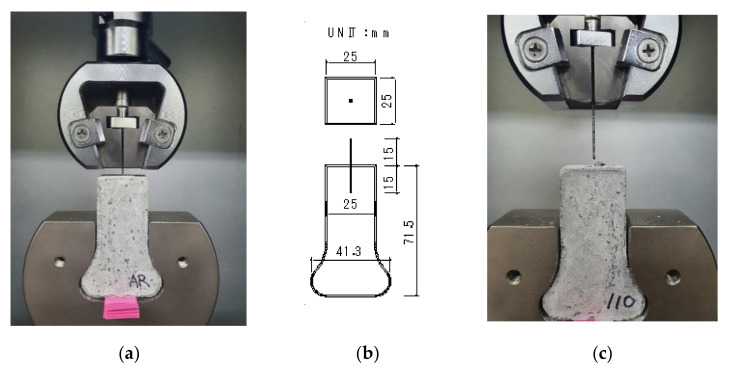
Test set-up and dimensions of a specimen: (**a**) test set-up; (**b**) dimensions of a specimen; and (**c**) photo of pulling out.

**Figure 12 materials-13-03845-f012:**
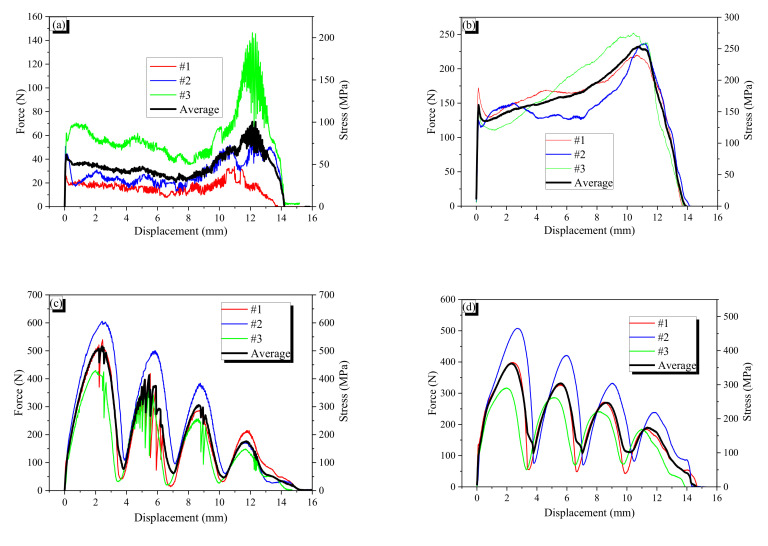
Force–displacement curves of the AR1.0 and CR1.0 fibers: (**a**) AR1.0-N fiber; (**b**) AR1.0-H fiber; (**c**) CR1.0-1N fiber; (**d**) CR1.0-1H fiber; (**e**) CR1.0-2N fiber; (**f**) CR1.0-2H fiber; (**g**) CR1.0-3N fiber; (**h**) CR1.0-3H fiber; (**i**) CR1.0-4N fiber; (**j**) CR1.0-4H fiber. The red, blue, green and black solid curves indicated the force-displacement curves of sample #1, #2, #3 and their average.

**Figure 13 materials-13-03845-f013:**
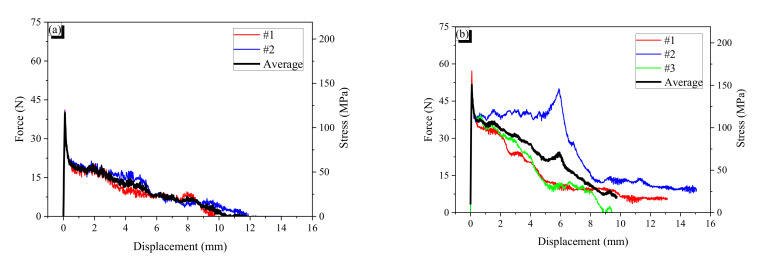
Force–displacement curves of the AR0.7 and CR0.7 fibers: (**a**) AR0.7-N fiber; (**b**) AR0.7-H fiber; (**c**) CR0.7-1N fiber; (**d**) CR0.7-1H fiber; (**e**) CR0.7-2N fiber; (**f**) CR0.7-2H fiber; (**g**) CR0.7-3N fiber; (**h**) CR0.7-3H fiber; (**i**) CR0.7-4N fiber; (**j**) CR0.7-4H fiber. The red, blue, green and black solid curves indicated the force-displacement curves of sample #1, #2, #3 and their average.

**Figure 14 materials-13-03845-f014:**
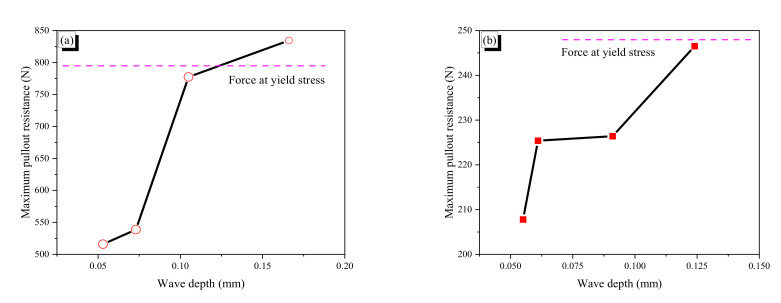
Maximum pullout resistance of the CR fibers: (**a**) CR1.0-N fibers; (**b**) CR0.7-N fibers.

**Figure 15 materials-13-03845-f015:**
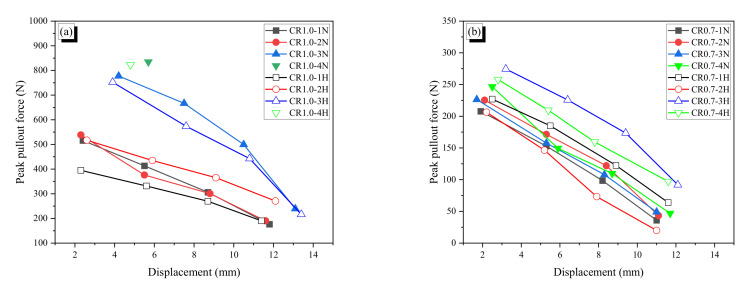
Peak pullout resistance with displacement: (**a**) CR1.0 fibers; (**b**) CR0.7 fibers.

**Figure 16 materials-13-03845-f016:**
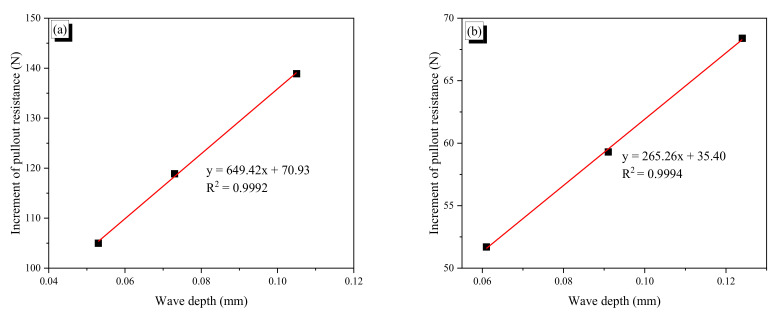
Increment of pullout resistance as a function of wave depth: (**a**) CR1.0-N fibers; (**b**) CR0.7-N fibers.

**Table 1 materials-13-03845-t001:** Reduction of diameter by cold drawing.

Fiber Types	AR1.0	AR0.7
Diameter (mm)	1.020	0.702
Reduction of diameter (mm)	0.065	0.042
ARR (%)	12.3	11.6

**Table 2 materials-13-03845-t002:** Dimensions of crimped fibers (unit: mm).

Name	Before Heating					After Heating				
	Thick- Ness	Height	Depth	WDR (%)	Length	Thick- Ness	Height	Depth	Length	∆L/L (%)
AR1.0	0.955	-	-		50.44	0.971	-	-	48.86	3.13
CR1.0-1	0.965	1.018	0.053	5.5	50.36	0.973	1.003	0.030	49.38	1.95
CR1.0-2	0.966	1.039	0.073	7.6	50.22	0.974	1.014	0.040	49.55	1.33
CR1.0-3	0.972	1.077	0.105	10.9	50.54	0.977	1.037	0.060	49.93	1.21
CR1.0-4	0.972	1.138	0.166	17.4	50.39	0.976	1.088	0.112	49.88	1.01
AR0.7	0.660	-	-		50.45	0.671	-	-	49.39	2.10
CR0.7-1	0.665	0.720	0.055	8.3	50.51	0.671	0.705	0.034	49.65	1.70
CR0.7-2	0.666	0.727	0.061	9.2	50.49	0.672	0.709	0.037	49.62	1.72
CR0.7-3	0.667	0.758	0.091	13.6	49.32	0.672	0.736	0.064	48.59	1.48
CR0.7-4	0.667	0.791	0.124	18.6	49.92	0.670	0.762	0.092	49.26	1.32

**Table 3 materials-13-03845-t003:** Yield stress of AR and CR fibers and the corresponding strain.

		AR1.0	CR1.0-1	CR1.0-2	CR1.0-3	CR1.0-4	AR0.7	CR0.7-1	CR0.7-2	CR0.7-3	CR0.7-4
Non-heating	Stress (MPa)	950	863	840	800	775	730	685	650	615	602
	Strain (%)	8.7	8.7	8.7	8.7	8.7	6.3	6.3	6.3	6.3	6.3
Heating	Stress (MPa)	979	1006	990	966	966	766	786	833	833	833
	Strain (%)	12.3	12.3	12.3	12.3	12.3	17.2	17.2	17.2	17.2	17.2

**Table 4 materials-13-03845-t004:** Maximum recovery stress and residual stress of AR and CR fibers.

Type	Maximum Recovery Stress (MPa)				Residual Stress (MPa)			
	100 °C	150 °C	200 °C	300 °C	100 °C	150 °C	200 °C	300 °C
AR1.0	222	317	363	370	105	190	217	246
CR1.0-1	151	269	255	270	64	138	153	188
CR1.0-2	143	235	224	244	61	113	130	168
CR1.0-3	111	199	172	192	48	106	112	138
CR1.0-4	72	137	134	141	24	82	94	115
AR0.7	200	257	264	269	127	176	196	232
CR0.7-1	156	145	200	209	91	113	169	197
CR0.7-2	152	203	232	229	74	139	175	208
CR0.7-3	128	132	203	212	72	96	163	190
CR0.7-4	135	174	168	227	74	121	140	209

**Table 5 materials-13-03845-t005:** Composition of the mortar.

Cement (Type III)	Fly Ash	Silica Sand	High-Range Water, Reducing Admixture	Water
1.00	0.15	1.00	0.009	0.35

**Table 6 materials-13-03845-t006:** Average peak force and response displacement of crimped fibers.

Type/ Peak Points	1^st^ Peak		2^nd^ Peak		3^rd^ Peak		4^th^ Peak		$\Delta_{1-2}$ (N)	$\Delta_{2-3}$ (N)	Average $\overset{-}{\Delta }$ (N)
	Force (N)	Slip (mm)	Force (N)	Slip (mm)	Force (N)	Slip (mm)	Force (N)	Slip (mm)			
CR1.0-1N	515.9	2.4	413.3	5.5	306.0	8.7	176.5	11.8	102.6	107.3	105.0
CR1.0-2N	538.6	2.3	376.1	5.5	300.8	8.8	190.4	11.6	162.5	75.3	118.9
CR1.0-3N	777.4	4.2	666.6	7.5	499.6	10.5	239.2	13.1	110.8	167.0	138.9
CR1.0-4N	834.5	5.7	-	-	-	-	-	-	-	-	-
CR1.0-1H	394.9	2.3	331.5	5.6	269.5	8.7	190.2	11.4	63.4	62.0	62.7
CR1.0-2H	517.1	2.6	435.0	5.9	365.1	9.1	270.7	12.1	82.1	69.9	76.0
CR1.0-3H	752.3	3.9	573.8	7.6	442.6	10.8	216.4	13.4	178.5	131.2	154.9
CR1.0-4H	823.2	4.8	-	-	-	-	-	-	-	-	-
CR0.7-1N	207.8	1.9	152.9	5.3	98.5	8.2	36.0	11.0	54.9	54.4	54.7
CR0.7-2N	225.4	2.1	171.4	5.3	122.0	8.4	43.4	11.1	54.0	49.4	51.7
CR0.7-3N	226.4	1.7	157.7	5.3	107.9	8.3	49.0	11.0	68.7	49.8	59.3
CR0.7-4N	246.5	2.5	149.4	5.9	109.7	8.7	46.8	11.7	97.1	39.7	68.4
CR0.7-1H	227.0	2.5	184.9	5.5	122.5	8.9	63.9	11.6	42.1	62.4	52.3
CR0.7-2H	206.4	2.2	146.5	5.2	73.3	7.9	20.0	11.0	59.9	73.2	66.6
CR0.7-3H	274.5	3.2	225.7	6.4	173.5	9.4	91.9	12.1	48.8	52.2	50.5
CR0.7-4H	257.9	2.8	209.5	5.4	159.5	7.8	97.2	11.6	48.4	50.0	49.2

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
