# Peer review of "Active Reinforcing Fiber of Cementitious Materials Using Crimped NiTi SMA Fiber for Crack-Bridging and Pullout Resistance"

_materials, 2020, doi:10.3390/ma13173845_

Round 1

Reviewer 1 Report

The paper presents a study of the mechanical properties and bond resistance of NiTi SMA fibers for use in cementitious materials.

Strengths:

  1. A very clear introduction and problem identification

Major comments:

  1. The structure of the paper is not well organised. I suggest applying a standard structure, e.g. 1. Introduction; 2. Materials and methods; 3. Results and discussion; 4. Conclusions. It would be easier to follow what materials and experimental setup were applied.
  2. What are the properties of the mortar used? What is, e.g. the type of cement, water to binder ratio, aggregate, etc.? Do the authors think that there is any influence of the interface “fiber-cement binder” on the fiber behaviour? Microstructural studies would give important complementary information on the mechanism.

Author Response

Reviewer #1:

The authors really thank you for your good comments.

  1. The structure of the paper is not well organized. I suggest applying a standard structure, e.g. 1. Introduction; 2. Materials and methods; 3. Results and discussion; 4. Conclusions. It would be easier to follow what materials and experimental setup were applied.

Response 1: The manuscript was re-structured based on the comment as below:

  1. Introduction
  2. Characteristics of crimped SMA fibers.

2.1 Properties of NiTi SMA fibers

       2.2 Production of crimped fibers

       2.3 Tensile behavior of the fibers

       2.4. Recovery stress of the crimped fiber

  1. Pullout test and results

3.1 Specimens and test set-up

       3.2 Pullout behavior

  1. Conclusions

  1. What are the properties of the mortar used? What is, e.g. the type of cement, water to binder ratio, aggregate, etc.? Do the authors think that there is any influence of the interface “fiber-cement binder” on the fiber behavior? Microstructural studies would give important complementary information on the mechanism.

Response 2: The properties of the mortar was added in the manuscript as below. The pullout resistance depended on the properties of fiber, of cement matrix and adhesion between them. Thus, the mortar’s properties affected on the pullout resistance. However, in this study, we focused on the effect of SMA fiber, especially the shape and initial diameter. The composition of mortar was selected based on a previous study (Choi, E., Kim, H. S., & Nam, T. H. (2020). Effect of crimped SMA fiber geometry on recovery stress and pullout resistance. Composite Structures, 112466.). There may need more studies to investigate the effect of mortar on the pullout resistance.

The properties of crimped fiber are presented above through the experimental tests. The compressive strength of the mortar is 55 MPa, and composition of the mortar is showed in Table 5.

Table 5. Composition of the mortar

Cement (type III)

Fly ash

Silica sand

High-range water, reducing admixture

Water

1.00

0.15

1.00

0.009

0.35

Reviewer 2 Report

The manuscript reports on an extensive experimental campaign on SMA fibers under various test methods. The manuscript is of good quality and suitable for publication after the following comments are addressed.

  • Overall, the level of English is good, but some phrases should be corrected, mainly in the introduction.
    • L26: “are good material”
    • L30: the choice of the term “chemical fibers” is rather strange, as this is definitely not a widespread or highly-used term. Perhaps another term would be better suited.
    • L44: “provide the only chemical bonds”
  • L48: “all of the fibers are inert in the high pH environment […] and have a high melting point”. While the former is true for most, glass fibers are especially prone to alkali attack. The latter (high melting point) is not true for the most commonly used polymeric fibers (PP, PE). These points should be added for clarity and completeness.
  • Nomenclature and ordering should be more consistent:
    • L87: “1.020 and 0.702 mm” while on L83 and L88 the order is reversed. Please correct to make it more consistent
    • Fig 7 and Table 4: naming is inconsistent (CR4-0.7 vs CR0.7-4)
  • Paragraph 2.1, L83-93 was not clear to the reviewer. Is it correct that the non-cold drawn fibers are just the ‘virgin’ fibers, while the production process for cold-drawn fibers is annealing-colddraw-heating-again cold draw? Please clarify.
  • L117: can the authors also include the “wave depth” in figure 3 for clarity?
  • Paragraph 2.3 lacks critical experimental info: imposed strain/stress rate, length of the fibers between the clamps. Furthermore, definitions of yielding and the elastic modulus (tangent or secant modulus?) should be given. Finally, references to test methods in literature should be provided, if available.
  • While the reviewer would like to complement the authors on the clear, and high resolution figures with clearly distinguishable lines, the calculation of the average in figure 11a (especially, but also some others) and figure 12 should be redone as to eliminate sudden jumps.
  • Additionally, some curiosities in some tests should be clarified explicitly.
    • For some tests, only 2 samples are available (AR07-N for example), and no explanation is given in the text.
    • A sudden decrease in AR1-H #3 is noticed. Why is this, is this real or maybe a measurement error?
    • The average for CR1-4N should maybe be split up between pull-out and failed fibers.
  • For figures 11, 12 and 13, the yield ‘stress’ of each fiber should be indicated as the authors refer to that yield stress (and its exceedance) multiple times in the text. Adding a horizontal line to Fig 11, 12 and 13 to indicate the ‘upper’ limit of the stress/force can improve understanding and readability.
  • The analysis on L313-315 seems arbitrary. The 3 points are fitted with a parabolic relation (which of course yields r² = 1), but this parabolic relation has no clear basis in reality or is of no physical meaning. Concluding then that the maximum force increases quadratically is of no importance, as the minimum of the parabola has no physical meaning either. Please clarify why a parabolic relation would be applicable here, rather than a linear or any other relation.
  • The discussion on L327 and following is not entirely fair either, as the diameter is dramatically different between both fiber types. Correcting for that and the different starting point, the proportional increase is more in line.
  • Related to the previous point, the reviewer does not agree with the point made on L335. While the moment of inertia is indeed smaller, the cantilever arm and (especially) the bending moment in the fiber will be smaller as well (due to the smaller pull-out force) for the 0.7 mm fiber. Therefore, the stress developed due to bending (according to the Navier equation) in the fiber could be similar between both fiber types and – given the same stiffness – the deformation would be roughly the same as well. In that light, it is not clear how the authors can explain the increment of the pull-out force based on the flexural rigidity. This point should be clarified further and in greater detail.

Overall, the manuscript can be accepted for publication if the above remarks are clarified and corrected.

Author Response

Reviewer #2:

  1. Overall, the level of English is good, but some phrases should be corrected, mainly in the introduction. L26: “are good material”

Response 1: The phrase “are good material” was edited to “are good materials” as below:

Cementitious materials are good materials for civil construction with low cost and convenience for shaping.

  1. L30: the choice of the term “chemical fibers” is rather strange, as this is definitely not a widespread or highly-used term. Perhaps another term would be better suited.

Response 2: The phrase “chemical fibers” was replaced by “synthetic fibers”:

Synthetic fibers are produced from petroleum or metallic, carbon, glass in a chemical process. The most common synthetic fibers used for cementitious materials are polypropylene, polyvinyl alcohol, glass, and carbon.

  1. L44: “provide the only chemical bonds”

Response 3: The phrase “only chemical bonds” was edited by “only chemical bond”

Polypropylene and polyvinyl alcohol are cheap and easy to mix, while they have a low modulus and provide the only chemical bond.

  1. L48: “all of the fibers are inert in the high pH environment […] and have a high melting point”. While the former is true for most, glass fibers are especially prone to alkali attack. The latter (high melting point) is not true for the most commonly used polymeric fibers (PP, PE). These points should be added for clarity and completeness.

Response 4: The sentence was re-written as below.

Some of the synthetic fibers such as polypropylene, polyvinyl alcohol are inert in the high pH environment of the cementitious matrix while carbon fibers have a high melting point; thus, synthetic fibers can minimize the harm of fire or environmental effect [19].

  1. Fig 7 and Table 4: naming is inconsistent (CR4-0.7 vs CR0.7-4)

Response 5: The “CR4-0.7” is in Figure 7j was edited to “CR0.7-4”.

  1. Nomenclature and ordering should be more consistent: L87: “1.020 and 0.702 mm” while on L83 and L88 the order is reversed. Please correct to make it more consistent. Paragraph 2.1, L83-93 was not clear to the reviewer. Is it correct that the non-cold drawn fibers are just the ‘virgin’ fibers, while the production process for cold-drawn fibers is annealing-cold draw-heating-again cold draw? Please clarify.

Response 6: Paragraph 2.1 was re-written. The “1.020 and 0.702 mm” diameter was indicated to calculate the reduction of cold drawing process. While the diameters of “virgin” fibers were 0.7 and 1.0 mm.

Ni50.4-Ti (wt.%) SMA fibers with two diameters of 0.7 and 1.0 mm were used for cold drawing and the subsequent fabrication of crimped fibers. First, they were annealed to make them straight and remove dislocation and then were cold down to room temperature. The cold drawing process started by heating the fibers with the temperature of 500℃, which made them bulged and the diameters increased to 1.020 mm and 0.702 mm. After that, the fibers were cold drawn under room temperature of about 25oC, and their final diameters became 0.955 mm and 0.660 mm, respectively. The reduction of diameters is summarized in Table 1.

Table 1: Reduction of diameter by cold drawing

Fiber types

AR1.0

AR0.7

Diameter (mm)

1.020

0.702

Reduction of diameter (mm)

0.065

0.042

ARR (%)

12.3

11.6

  1. L117: can the authors also include the “wave depth” in figure 3 for clarity?

Response 7: The “wave depth” was added in Figure 3.

  • Crimping process (b) Shape of a crimped fiber       

(c) Photos of AR and CR fibers

Figure 3. Crimping process and shape of crimped fiber

  1. Paragraph 2.3 lacks critical experimental info: imposed strain/stress rate, length of the fibers between the clamps. Furthermore, definitions of yielding and the elastic modulus (tangent or secant modulus?) should be given. Finally, references to test methods in literature should be provided, if available.

Response 8: The information of tensile test in paragraph 2.3 was describe as below. The test set-up was based on a previous study [Choi, E., Ostadrahimi, A., & Park, J. (2020). On mechanical properties of NiTi SMA wires prestrained by cold rolling. Smart Materials and Structures29(6), 065009]. The definitions of modulus and yielding were also added.

Tensile tests were conducted with two different initial diameters of the AR fibers as well as CR fibers, namely, AR1.0, AR0.7, CR1.0-1 to -4, and CR0.7-1 to -4. The tests were controlled by displacement control with a loading speed of 1.0 mm/min. under room temperature of approximately of 25 oC. A fiber with a length of 50 mm was gripped by two cramps with 5 mm from each end of fiber, and test set-up was presented in Figure 5.

                              Figure 5: Monotonic tensile test setup

Without heating treatment, the cold drawn fibers did not show phase transformation. The stress-strain curves of the AR1.0-N fiber showed the initial linear part which increased up to 1.0% with the secant modulus of 22.6 GPa, and the following part which showed softening behavior with a secant modulus of 12.0 GPa until yielding at the strain of 8.7%. After the yield, the AR1.0-N showed plastic behavior until failure. The AR1.0-N fiber and CR1.0-N fibers showed almost the same tensile behavior up to 1.0% strain. Then, the crimped fibers showed softening behavior more clearly than AR fiber due to the stretching effect. At the strain of 8.7%, the AR fiber yielded at 950 MPa while the CR1.0-1N to CR1.0-4N fibers had the stresses of 863 MPa, 840 MPa, 800 MPa, and 775 MPa, respectively. The crimped fiber with a larger wave height showed lower stress. The crimped fibers showed continuously increased stresses from 9% to 18% before yielding. The Young’s modulus of SMA fibers does not depend on their diameter; the elastic modulus of the 0.7 diameter fibers was 22.6 MPa at the strain of 1.0%. However, after the initial elastic part, the AR0.7-N fiber had a secant modulus of 10.5 GPa, which was lower than the AR1.0-N fiber’ modulus.

  1. While the reviewer would like to complement the authors on the clear, and high-resolution figures with clearly distinguishable lines, the calculation of the average in figure 11a (especially, but also some others) and figure 12 should be redone as to eliminate sudden jumps.

Response 19: The type and width of lines in Figure 11, 12 and 8, 9 were edited to be more clearly.

  1. Additionally, some curiosities in some tests should be clarified explicitly. For some tests, only 2 samples are available (AR07-N for example), and no explanation is given in the text.

Response 10: For pullout test of AR07-N, the force-displacement curve of the third specimen was totally different with that of the first and second specimens. It showed the plateau force at the end period may be due to the sharp end cut of the fiber. Thus, this curve was excluded. For CR1.0-3N, a sample was not gridded by the clamps due to the mistake of making test.

  1. A sudden decrease in AR1-H #3 is noticed. Why is this, is this real or maybe a measurement error?

Response 11: The sudden decrease in AR1-H#3 was a measurement error; the sudden decrease only happened with this specimen while #1 and #2 was not. Moreover, all specimens of the AR1.0 had the sharp end cut, which made the error in end part of curves. Thus, the AR1-H#3 as well as AR1-H#1, AR1-H#2 was used to indicate the initial average curves. Thus, the sudden dropped data are removed from the graph.

  1. The average for CR1-4N should maybe be split up between pull-out and failed fibers.

Response 12: Based on the comment, the average is calculated again using only the data of #2 and #3; thus, the data of #1 is exclude form the average.

  1. For figures 12, 13 and 14, the yield ‘stress’ of each fiber should be indicated as the authors refer to that yield stress (and its exceedance) multiple times in the text. Adding a horizontal line to Fig 12, 13 and 14 to indicate the ‘upper’ limit of the stress/force can improve understanding and readability.

Response 13: The “upper limit” line of yield stress was added in the manuscript in Figure 12 g-j, figure 13 g-j, and figure 14.

  1. The analysis on L313-315 seems arbitrary. The 3 points are fitted with a parabolic relation (which of course yields r² = 1), but this parabolic relation has no clear basis in reality or is of no physical meaning. Concluding then that the maximum force increases quadratically is of no importance, as the minimum of the parabola has no physical meaning either. Please clarify why a parabolic relation would be applicable here, rather than a linear or any other relation.

Response 14: The discussion about the parabolic relation was deleted.

  1. The discussion on L327 and following is not entirely fair either, as the diameter is dramatically different between both fiber types. Correcting for that and the different starting point, the proportional increase is more in line.
  2. Related to the previous point, the reviewer does not agree with the point made on L335. While the moment of inertia is indeed smaller, the cantilever arm and (especially) the bending moment in the fiber will be smaller as well (due to the smaller pull-out force) for the 0.7 mm fiber. Therefore, the stress developed due to bending (according to the Navier equation) in the fiber could be similar between both fiber types and – given the same stiffness – the deformation would be roughly the same as well. In that light, it is not clear how the authors can explain the increment of the pull-out force based on the flexural rigidity. This point should be clarified further and in greater detail.

Response 15& 16: The authors thank the good comments. However, some points should be mentioned. The cantilever arm, which is related to the wave depth of a crimped fiber, of the 0.7 mm fiber is always not smaller than that of the 1.0 mm fiber. For example, the CR0.7-4 has a wave depth of 0.124 mm, which is similar to 0.105 mm wave depth of the CR1.0-3. The bending moment is related to the applied pullout force, and thus it can not be the property of a CR fiber. The reviewer mentioned that the stress and deformation would be similar if the stiffness is same; the authors agree with this comment. In the paper, the authors assumed the crimped fibers of 0.7 mm and 1.0 mm have the same cantilever length, namely wave depth, and then the flexural rigidity controls the deformation due to bending. Therefore, the flexural rigidity is correct on only the assumption that both types of fibers have the same cantilever length. Thus, the authors tried to revise the related sentence like below to used flexural stiffness instead of the flexural rigidity:

This phenomenon is highly related to flexural stiffness of the bent part in a crimped fiber. The flexural stiffness of the bent part is function of wave depth, which acts like a cantilever arm in bending, and flexural rigidity (EI) of the fiber. Thus, if both types of CR0.7-N and CR1.0-N fibers have the same wave depth, the flexural stiffness of the CR1.0-N fiber have four times larger than that of the CR0.7-N fiber. Therefore, it can be said for the crimped SMA fiber that the maximum pullout force is highly related to the flexural stiffness of the bent part.

Reviewer 3 Report

The work presented in the manuscript is to investigate the mechanical behavior of crimped SMA fiber. It would be a valuable research for the future application of SMA fibers in cementitious materials. Though I value the research significance, I urge authors to add more explanation on the experimental program and test results, the interpretation of the test results, and elaborate the figures. 

Here are the general comments.

  • Line 31~32: steel fiber produced various shape to enhance the anchorage and bond properties as such it improves the energy dissipation. The cheap price is not the reason for various shape.
  • Line 41~50: the behavior and purpose of chemical fiber is not comparable with steel fiber and SMA fiber. In my opinion, chemical fiber is irrelevant in the context of the paper. I propose deleting this paragraph.
  • Line 109-132: Authors may need to revisit the definition of crimped SMA fiber dimensions
    • Wave height: The wave height should be measured at the centroid of the steel fiber excluding the thickness of the fiber. Is this a conventional way of defining the wave height?
    • “wave depth?”. I think the authors would like to describe the crimping depth.
    • in some cases, wave height and wave depth were interchanged. However, based on the definition written in the manuscript, those are different. 
  • Line 133-145: What was the heating protocol to measure the length shrinking ratio?
  • Line 165-181: it is interesting that the initial tensile behaviors of AR and CR fiber are similar. And the stretching effect of crimped fibers kicks in after 250 MPa.
  • Line 165-192: What method used to identify the yield strength of fiber? Is there any standard test method available for a steel fiber? Please explain briefly how authors determined the yield strength.
  • Section 3.1. Test Set-up: Please clarify the heating protocol.
    • The type k thermocouple was wrapped around a SMA fiber. How could you ensure the temperature of the thermocouple and the fiber were the same? Did the authors apply hot gun at a certain degree for a few minutes to ensure the fiber temperature equilibrated with the applied temperature? Or the heat application was stopped when the thermocouple reached at a target temperature? If the latter case is what authors did, it explains incomplete phase transformation at 100 degree.
  • Though the compressive strength of mortar has not been a major variable that authors investigated, it should be reported in the main body of manuscript.

Authors can find a further comments from the attached file. 

Author Response

Reviewer #3:

  1. The Line 31~32: steel fiber produced various shape to enhance the anchorage and bond properties as such it improves the energy dissipation. The cheap price is not the reason for various shape.

Response 1:  The authors agree with the reviewer. That is our mistake with the English. We fixed the sentence as below:

Steel fibers are cheap and provide enough energy dissipation after cracking; they are commonly used with several shapes, such as end-hooked, twisted, and crimped [10,11].

  1. Line 41~50: the behavior and purpose of chemical fiber is not comparable with steel fiber and SMA fiber. In my opinion, chemical fiber is irrelevant in the context of the paper. I propose deleting this paragraph.

Response 2:  The SMA fiber has some advantages against the chemical (synthetic) fiber, which is   inert in the high pH environment of the cementitious matrix and has a high melting point. Moreover, the chemical fiber is widely used in the cementitious materials in construction. Thus, the authors want to mention the fiber in the introduction part.

  1. Line 109-132: Authors may need to revisit the definition of crimped SMA fiber dimensions.

3.1. Wave height: The wave height should be measured at the centroid of the steel fiber excluding the thickness of the fiber. Is this a conventional way of defining the wave height?

3.2. “wave depth?”. I think the authors would like to describe the crimping depth.

3.3. in some cases, wave height and wave depth were interchanged. However, based on the definition written in the manuscript, those are different. 

Response 3:  The shape of a crimped fiber and its dimensions are illustrated in Figure 3 to make the definition more clearly. If thickness of the fiber is fixed, wave depth is exactly related to wave height, and both parameters can highly effect on the bond resistance. Thus, they can be used to explain the influence on the bond resistance.

The wave height was measured along wave direction, and the thickness was measured perpendicular to the wave. The difference between these two values is the wave depth.

  • Crimping process (b) Shape of a crimped fiber

(c) Photos of AR and CR fibers

Figure 3. Crimping process and shape of crimped fiber

  1. Line 133-145: What was the heating protocol to measure the length shrinking ratio?

Response 4:  The fibers in the tests were heated with flame to increase temperature quickly resulting in the phase transformation. Thus, the related sentence is revised like below:

When the fibers are heated with flame without any restraint, an AR fiber shrinks because of deformation recovery due to the shape memory effect.

  1. Line 165-181: it is interesting that the initial tensile behaviors of AR and CR fiber are similar. And the stretching effect of crimped fibers kicks in after 250 MPa.

Response 5:  The stress-strain curves of the AR and CR fibers showed the same initial linear part up to 1.0% with the slope of 22.6 GPa. With initial low stress less than 250 MPa, the bending moment was too small to stretch the bent part because the flexural stiffness of the crimped fiber is enough to bear the bending moment; thus, the indentation was not be stretched.

  1. Line 165-192: What method used to identify the yield strength of fiber? Is there any standard test method available for a steel fiber? Please explain briefly how authors determined the yield strength.

Response 6:  The yield strength was indicated at the point where the curve started the decrement of slope; the fiber showed the plastic behavior after this point. The graph was re-written as below:

Without heating treatment, the cold drawn fibers did not show phase transformation. The stress-strain curves of the AR1.0-N fiber showed the initial linear part which increased up to 1.0% with the secant modulus of 22.6 GPa, and the following part which showed softening behavior with a secant modulus of 12.0 GPa until yielding at the strain of 8.7%. After the yield, the AR1.0-N showed plastic behavior until failure. The AR1.0-N fiber and CR1.0-N fibers showed almost the same tensile behavior up to 1.0% strain. Then, the crimped fibers showed softening behavior more clearly than AR fiber due to the stretching effect. At the strain of 8.7%, the AR fiber yielded at 950 MPa while the CR1.0-1N to CR1.0-4N fibers had the stresses of 863 MPa, 840 MPa, 800 MPa, and 775 MPa, respectively. The crimped fiber with a larger wave height showed lower stress. The crimped fibers showed continuously increased stresses from 9% to 18% before yielding. The Young’s modulus of SMA fibers does not depend on their diameter; the elastic modulus of the 0.7 diameter fibers was 22.6 MPa at the strain of 1.0%. However, after the initial elastic part, the AR0.7-N fiber had a secant modulus of 10.5 GPa, which was lower than the AR1.0-N fiber’ modulus.

  1. Section 3.1. Test Set-up: Please clarify the heating protocol. The type k thermocouple was wrapped around a SMA fiber. How could you ensure the temperature of the thermocouple and the fiber were the same? Did the authors apply hot gun at a certain degree for a few minutes to ensure the fiber temperature equilibrated with the applied temperature? Or the heat application was stopped when the thermocouple reached at a target temperature? If the latter case is what authors did, it explains incomplete phase transformation at 100 degree.

Response 8:  The authors totally agree with your opinion and really appreciate great comments. Thus, the below paragraph is added in the end part of the section.

In the tests, it should be notified that the temperature of type k thermocouple is exactly same as that of the heated fiber because the heating was stopped when the temperature of the type k thermocouple reached the target value; thus, the temperature of the fiber may be lower than that of the type k thermocouple. For the heating cases with 100℃, the fibers may be not experienced complete phase transformation. If a whole fiber undergoes the phase transformation, the recovery stress becomes stable and decreases with more increasing temperature because of thermal expansion of the fiber; the decrement in the recovery stress is observed for the heating cases of 200℃ or 300℃.

  1. Though the compressive strength of mortar has not been a major variable that authors investigated, it should be reported in the main body of manuscript.

Response 9:  The properties of the mortar was added in the manuscript as below.

The properties of crimped fiber are presented above through the experimental tests. The compressive strength of the mortar is 55 MPa, and composition of the mortar is showed in Table 5.

Table 5. Composition of the mortar

Cement (type III)

Fly ash

Silica sand

High-range water, reducing admixture

Water

1.00

0.15

1.00

0.009

0.35

Reviewer 4 Report

The paper presents very interesting test results of a few types of Ni-Ti crimped SMA fibres compared with straight fibres. Description of all tests and test results is very detailed and comprehensible. The only critical comment I have is that it should be explain more clearly what means “free heating conditions” mentioned in chapter 2.2 and referred to later in the paper when heated and not heated specimens are compered.  I also miss information about number of specimen tested (except pull out tests).

Author Response

Reviewer #4:

  1. The paper presents very interesting test results of a few types of Ni-Ti crimped SMA fibres compared with straight fibres. Description of all tests and test results is very detailed and comprehensible. The only critical comment I have is that it should be explain more clearly what means “free heating conditions” mentioned in chapter 2.2 and referred to later in the paper when heated and not heated specimens are compered.

Response 1: The above reviewer provided the same comment about the free heating conditions, and thus the sentence is revised like below:

When heated without any restraint, an AR fiber shrinks because of deformation recovery due to the shape memory effect.

  1. I also miss information about number of specimens tested (except pull out tests).

Response 2: For tensile tests and recovery tests, only one specimen was used. Meanwhile, if a result is deviated from the expectation, then the result was thrown away and the test was conducted again. Thus, the data seems to be reliable.

Reviewer 5 Report

Positive properties:

The topic is interesting and original. The authors fond relations between pullout resistance of the crimped fiber and the wave depth, embedded length, yield strength, and bending rigidity.

General notes:

Pullout resistance of crimped SMA fibers could depend on composition of cement mortar. It is necessary to show the composition in more detail: aggregate maximal diameter, micro-filler content and water-cement ratio.

The fibers are designed for reinforcing cementitious materials. It would be good to mention some practical examples of fiber application in construction members.

Author Response

Reviewer #5:

  1. The topic is interesting and original. The authors fond relations between pullout resistance of the crimped fiber and the wave depth, embedded length, yield strength, and bending rigidity.

Response 1: The authors really appreciate your positive comments.

  1. Pullout resistance of crimped SMA fibers could depend on composition of cement mortar. It is necessary to show the composition in more detail: aggregate maximal diameter, micro-filler content and water-cement ratio.

Response 2: The composition of mortar is provided like below:

The properties of crimped fiber are presented above through the experimental tests. The compressive strength of the mortar is 55 MPa, and composition of the mortar is showed in Table 5.

Table 5. Composition of the mortar

Cement (type III)

Fly ash

Silica sand

High-range water, reducing admixture

Water

1.00

0.15

1.00

0.009

0.35

  1. The fibers are designed for reinforcing cementitious materials. It would be good to mention some practical examples of fiber application in construction members.

Response 2: Thank you for the good comment. As mentioned in the introduction, the SMA fibers can be used for crack-repairing, delaying cracking, and crack-bridging. Thus, the SMA fibers can be distributed in the cover concrete; this method can be applied for any type of concrete member. Now, the authors are trying to apply the SMA fiber on reinforced concrete column, in which cracks are developed at around plastic region due to bending. Thus, the SMA fiber is expected to repair cracks in RC columns.

Round 2

Reviewer 1 Report

The paper has been revised and improved. All the comments were addressed. It can be accepted in current form.

Author Response

Dear Reviewer:

Thank you for your comment.

Kindly regards,

Choi.

Reviewer 2 Report

no further comments

Author Response

(The authors gave the same response as above.)

Reviewer 3 Report

Thank you for the authors to revise the manuscript and clarify the general comments that I made. However, I wonder if the authors have opened the file attached in the first round of review. It has additional detailed comments in the manuscript. And I do not see any changes regarding the additional comments I made in that file. I am attaching the file again. Please made corrections or clarify the manuscript or provide justifications. I cannot accept the publication because the comments have not been addressed in the manuscript. 

Thanks,

Author Response

Dear the reviewer:

Last time, the authors revised the paper sincerely following your comments. However, I am not familiar with the system, and then I uploaded wrong information. Thus, I added the response and revision.

I really appreciate your comments.

Kindly regards,

Choi.

Reviewer #3:

  1. The Line 31~32: steel fiber produced various shape to enhance the anchorage and bond properties as such it improves the energy dissipation. The cheap price is not the reason for various shape.

Response 1:  The authors agree with the reviewer. That is our mistake with the English. We fixed the sentence as below:

Steel fibers are cheap and provide enough energy dissipation after cracking; they are commonly used with several shapes, such as end-hooked, twisted, and crimped [10,11].

  1. Line 41~50: the behavior and purpose of chemical fiber is not comparable with steel fiber and SMA fiber. In my opinion, chemical fiber is irrelevant in the context of the paper. I propose deleting this paragraph.

Response 2:  The SMA fiber has some advantages against the chemical (synthetic) fiber, which is   inert in the high pH environment of the cementitious matrix and has a high melting point. Moreover, the chemical fiber is widely used in the cementitious materials in construction. Thus, the authors want to mention the fiber in the introduction part.

  1. Line 109-132: Authors may need to revisit the definition of crimped SMA fiber dimensions.

3.1. Wave height: The wave height should be measured at the centroid of the steel fiber excluding the thickness of the fiber. Is this a conventional way of defining the wave height?

3.2. “wave depth?”. I think the authors would like to describe the crimping depth.

3.3. in some cases, wave height and wave depth were interchanged. However, based on the definition written in the manuscript, those are different. 

Response 3:  The shape of a crimped fiber and its dimensions are illustrated in Figure 3 to make the definition more clearly. If thickness of the fiber is fixed, wave depth is exactly related to wave height, and both parameters can highly effect on the bond resistance. Thus, they can be used to explain the influence on the bond resistance.

The wave height was measured along wave direction, and the thickness was measured perpendicular to the wave. The difference between these two values is the wave depth.

  • Crimping process (b) Shape of a crimped fiber

(c) Photos of AR and CR fibers

Figure 3. Crimping process and shape of crimped fiber

  1. Line 133-145: What was the heating protocol to measure the length shrinking ratio?

Response 4:  The fibers in the tests were heated with flame to increase temperature quickly resulting in the phase transformation. Thus, the related sentence is revised like below:

When the fibers are heated with flame without any restraint, an AR fiber shrinks because of deformation recovery due to the shape memory effect.

  1. Line 165-181: it is interesting that the initial tensile behaviors of AR and CR fiber are similar. And the stretching effect of crimped fibers kicks in after 250 MPa.

Response 5:  The stress-strain curves of the AR and CR fibers showed the same initial linear part up to 1.0% with the slope of 22.6 GPa. With initial low stress less than 250 MPa, the bending moment was too small to stretch the bent part because the flexural stiffness of the crimped fiber is enough to bear the bending moment; thus, the indentation was not be stretched.

  1. Line 165-192: What method used to identify the yield strength of fiber? Is there any standard test method available for a steel fiber? Please explain briefly how authors determined the yield strength.

Response 6:  The yield strength was indicated at the point where the curve started the decrement of slope; the fiber showed the plastic behavior after this point. The graph was re-written as below:

Without heating treatment, the cold drawn fibers did not show phase transformation. The stress-strain curves of the AR1.0-N fiber showed the initial linear part which increased up to 1.0% with the secant modulus of 22.6 GPa, and the following part which showed softening behavior with a secant modulus of 12.0 GPa until yielding at the strain of 8.7%. After the yield, the AR1.0-N showed plastic behavior until failure. The AR1.0-N fiber and CR1.0-N fibers showed almost the same tensile behavior up to 1.0% strain. Then, the crimped fibers showed softening behavior more clearly than AR fiber due to the stretching effect. At the strain of 8.7%, the AR fiber yielded at 950 MPa while the CR1.0-1N to CR1.0-4N fibers had the stresses of 863 MPa, 840 MPa, 800 MPa, and 775 MPa, respectively. The crimped fiber with a larger wave height showed lower stress. The crimped fibers showed continuously increased stresses from 9% to 18% before yielding. The Young’s modulus of SMA fibers does not depend on their diameter; the elastic modulus of the 0.7 diameter fibers was 22.6 MPa at the strain of 1.0%. However, after the initial elastic part, the AR0.7-N fiber had a secant modulus of 10.5 GPa, which was lower than the AR1.0-N fiber’ modulus.

  1. Section 3.1. Test Set-up: Please clarify the heating protocol. The type k thermocouple was wrapped around a SMA fiber. How could you ensure the temperature of the thermocouple and the fiber were the same? Did the authors apply hot gun at a certain degree for a few minutes to ensure the fiber temperature equilibrated with the applied temperature? Or the heat application was stopped when the thermocouple reached at a target temperature? If the latter case is what authors did, it explains incomplete phase transformation at 100 degree.

Response 8:  The authors totally agree with your opinion and really appreciate great comments. Thus, the below paragraph is added in the end part of the section.

In the tests, it should be notified that the temperature of type k thermocouple is exactly same as that of the heated fiber because the heating was stopped when the temperature of the type k thermocouple reached the target value; thus, the temperature of the fiber may be lower than that of the type k thermocouple. For the heating cases with 100℃, the fibers may be not experienced complete phase transformation. If a whole fiber undergoes the phase transformation, the recovery stress becomes stable and decreases with more increasing temperature because of thermal expansion of the fiber; the decrement in the recovery stress is observed for the heating cases of 200℃ or 300℃.

  1. Though the compressive strength of mortar has not been a major variable that authors investigated, it should be reported in the main body of manuscript.

Response 9:  The properties of the mortar was added in the manuscript as below.

The properties of crimped fiber are presented above through the experimental tests. The compressive strength of the mortar is 55 MPa, and composition of the mortar is showed in Table 5.

Table 5. Composition of the mortar

Cement (type III)

Fly ash

Silica sand

High-range water, reducing admixture

Water

1.00

0.15

1.00

0.009

0.35

Round 3

Reviewer 3 Report

I still do not see the responses to the comments that I made in the manuscript file. I guess authors have a difficulty of opening the attached file. Here are the comments I made in the manuscript version 1. Please revise or answer my comments before the final submission. 

Line 77: … mass production ability [40] ⇒ producibility

Line 87-88: The final diameters of the fibers after cold drawing because 0.966 mm and 0.660 mm, respectively. ⇒ Isn’t this diameter after the 500-degree heat treatment?

Line 90: non-cold drawn SMA fiber ⇒ is the non-cold drawn SMA fiber is the fiber prior to the cold drawn procedure? Or the non-cold drawn SMA fiber has different procedure?

Line 96: … differential scanning calorimetry (DSC) curves… ⇒ add one or two sentences explaining how DSC can capture the SMA phase transformation.

Line 109: … a special device called a rolling device… ⇒ just “a rolling device”

Line 110: …two confronting gears… ⇒ awkward expression. How about “A pair of crimper gears”?

Line 112: …25 mm rotate around their centers… ⇒ …25 mm rotate about their centers…

Line 112-113: …and press the as-received SMA wires that are passing through the rolling gears. ⇒ …crimp SMA fibers as they are fed through the rolling gears.

Line 115: the small gap ⇒ a small gap

Line 121-122: The crimped fiber became thicker than the AR fiber due to the crimping because of Poisson’s effect. ⇒ I believe this might be only true where the fiber was crimped. Does the thickness of crimped fiber measure at the thickest part of crimped fiber?

Line 133: because of, due to used sequentially. Not read well. How about "because of the deformation recovery effect of shape memory alloy (SMA)"

Line 137-138: …Table 2, while the values of the CR1.0 fibers smaller than the AR1.0 fiber 137 decreased with more indentation, … ⇒ I cannot understand what this means. Please explain clearly.

Line 143-145: The deformation recovery and stretching effect of the CR fiber 143 strongly affected development of recovery stress, and geometrical variation of wave depth and 144 thickness influenced bond resistance of the CR fiber. ⇒ This is why authors tested deformation recovery behavior of AR and CR fibers. With that said, this sentence needs to move at the beginning of this paragraph.

Table 2: TH ⇒ if this is an abbreviation of thickness, write notes under the table.

Figure 4 (b): typo. Dercement of wave depth ⇒ decrement of wave depth

Line 157: …conducted with four types of… ⇒ conducted on four types of

Line 157-158: …of cold drawn fibers as well as the AR fibers,… ⇒ Did authors would like to write "crimped and non-crimped fibers"?

Line 166: AR1.0-N ⇒ this is AR fiber without heat treatment please explain the nomenclature before using it.

Line 169: softening behavior ⇒ In this case, “lesser stiffness than AR fiber due to the stretching effect of crimped fibers.”

Line 173: 9% to 18% ⇒ is this of yield strain or yield stress?

Line 174-175: the elastic modulus of the 0.7 diameter fibers was 22.6 MPa at the strain of 1.0%. ⇒ this does not support the previous sentence.

Line 176: … than the AR1.0-N fiber’ modulus. ⇒ what was the modulus of AR1.0-C fiber?

Line 182: The response strain… ⇒ The strain…

Line 185-186: … is more effective than the smaller one. ⇒ decrement of ultimate strain is better in terms of behavior? that is why authors used "effective" term? "Effective" has positive meaning. Is the smaller ultimate strain better for the crack bridging behavior of cementitious materials with SMA fiber?

Line 201: k-coupler ⇒ K-coupler sounds like a product name. I guess authors refer to type K thermocouple. If k-coupler is something specific type of temperature gauge, please explain it. Throughout the manuscript.

Line 216: It is conjectured that the fibers… ⇒ awkward. "deemed" instead?

Line 217-219: … the Af (91.46 °C) where the phase transformation to austenite is completed. Therefore, it seems that all parts of the fiber may not have undergone phase transformation. ⇒ conflicting. "phase transformation is completed" therefore "not have undergone phase transformation". It was difficult to understand this. Maybe the heat was not well transferred to the fiber as such the phase transformation was not completed.

Line 236-237: For the prestressing effect, the fiber could be heated with a higher temperature to have more benefits. ⇒ This contradict the previous sentence.

Line 246-247: …values; this is totally different from the trend of the CR1.0 fibers. ⇒ Can authors explain why CR0.7-2 showed the highest values?

Figure 7: recommend different line types for each temperature plot

- (j): specimen ID – typo à (j) CR0.7-4

- Use a consistent font size for figure legends, axis tick values

Line 269-270: The temperature was increased rapidly in the beginning, and the increment became blunt after one hour and became stable at 145 °C in three hours. ⇒ “at” , also please revise this sentence.

Line 277: … and the fiber was kept with a 5 mm length from the top surface of the mortar specimen by an actuator. ⇒ Sentence doesn't make sense. Was the fiber gripped by wedge grips?

Line 335: … CR0.7-N fiber in the duct may not… ⇒ what does duct indicate?

Line 402: Crimped fibers show lower recovery stress and residual stress in comparison with AR fiber having 402 the same initial diameter because of the stretching effect. ⇒ when heat treated.

Line 421: The compressive strength of the mortar also can… ⇒ Authors should report measured compressive strength of the mortar in the experimental program section.

Thanks,

Author Response

Dear Reviewer 3:

I really appreciate your concerns and effort to complete the review. Now, I understand the system and revised the paper following your comments.

Kindly regards,

Choi

Reviewer #3: Additional comments

  1. page 1: various shapesà several shapes
  2. page 2: production ability à producibility
  3. page 2:

The related sentence of “The final diameter~~” is revised like below:

The cold drawing process started by heating the fibers with the temperature of 500℃, which made them bulged and the diameters increased to 1.020 mm and 0.702 mm. After that, the fibers were cold drawn under room temperature of about 25oC, and their final diameters became 0.955 mm and 0.660 mm, respectively.

  1. non-cold drawn SMA fiber -à non-cold drawn SMA fiber, which are prior to the cold drawing.

  1. page 3: About the DCS, the below sentence is added:

The DSC is a thermoanalytical technique, where the difference in the heating amount required to increase the temperature of a sample and reference is measured as a function of temperature

  1. page 3: About figure 1, the figure is revised following the comment like below:

Figure 1. DSC curves of SMA fibers before and after cold drawing

  1. page 3: a special device called a rolling deviceà The related sentence is revised.

The crimper gears with an outside diameter of 26 mm rotate about their centers and crimp SMA wires as they are fed through the rolling gears.

  1. page 4: The crimped fiber became thicker than the AR fiber due to the crimping because of Poisson’s effect.

Comment: I believe this might be only true where the fiber was crimped. Does the thickness of crimped fiber measured at the thickest part of crimped fiber.

Response: The reviewer’s opinion is correct. The thickness of the crimped fiber is measured at the bulged part. The perpendicular part of the bulged part is crimped and becomes saw-toothed, and, thus the thickness in the direction is not measured. Based on the comment, a sentence is added.

The crimped fiber became thicker than the AR fiber due to the crimping because of Poisson’s effect; the thickness is measured at the thickest part perpendicular to wave direction.

  1. page 4: under free condition à The related sentence is revised like below:

When the fibers are heated with flame without any restraint, an AR fiber shrinks because of deformation recovery of the shape memory effect.

  1. page 4: while the values of the CR1.0 fibers smaller than the AR1.0 fiber decreased with more indentation. --> while the values of the CR1.0 fibers, which are smaller than that of the AR1.0 fiber, decreases with more indentation,
  2. page 4: are authors comparing between CR0.7-1 and CR0.7-2?

Response: Yes, it is. The sentence is revised to avoid confusing.

the CR0.7-1 and CR0.7-2

  1. page 4: This is why authors tested deformation recovery behavior of AR and CR fibers. Then, this sentence needs to move at the beginning of this paragraph.

Response: The sentence is moved at the beginning of the paragraph.

  1. page 4: in figure 2, the (a) and (b) are inserted:

  (a) Rolling device                            (b) Dimension of the gear

Figure 2. Rolling device with the dimension of the gear

  1. page 5: figure numbering in figure 3:

  • Crimping process (b) Shape of a crimped fiber

(c) Photos of AR and CR fibers

Figure 3. Crimping process and shape of crimped fiber

  1. page 5: TH-à Thickness

  1. page 5: Dercement in figure 4 à. Decrement
  2. page 5: The captions in the graphs indicate the height of each column.

18.Page 6: The related part is revised like below:

Tensile tests were conducted on the AR fibers with two different initial diameters as well as on the CR fibers, namely, AR1.0, AR0.7, CR1.0-1 to -4, and CR0.7-1 to -4. The tests were controlled by displacement control with a loading speed of 1.0 mm/min. under room temperature of approximately of 25 oC. A fiber with a length of 50 mm was gripped by two cramps with 5 mm from each end of fiber, and test set-up was presented in Figure 5.

  1. page 6: About the AR1.0-N, the sentence is added.

the AR1.0-N fiber, where ‘-N’ indicates the case without heat treatment,

  1. page 6: softening behavior à the sentence is revised like below:

Then, the crimped fibers showed lesser stiffness more clearly than AR fiber due to the stretching effect at the crimped part.

  1. page 6: 9% to 18% à This indicates the increment of stress of the CR fibers.

  1. page 6: the elastic modulus of the 0.7 diameter fibers was~~~ à The related sentences are revised:

The secant modulus of the 0.7 diameter fibers was 22.6 MPa at the strain of 1.0%, which is equal to that of the AR1.0-N fiber. However, after the initial elastic part, the AR0.7-N fiber had a secant modulus of 10.5 GPa, which is lower than the modulus of the AR1.0-N fiber.

  1. page 6: ‘response’ is deleted.

  1. page 6: the larger diameter crimped fiber is more effective than the smaller one. à the sentence is removed.

  1. page 7:

Comment: How were these values determined? Are those corresponding to the second slope change? The question is what method the authors used for defining the yielding point for each fiber. If crimped fiber has special yield point definition, explain it explicitly and discuss the difference between straight and crimped fibers' yield points.

Response:

  1. page 7: k-coupler is changed to type k thermocouple.

A type k thermocouple, which is blue electrical wire in Figure 7, was wound on the fiber surface to check the temperature variation

  1. page 7: Temperature of 150℃, however, ~~ à the sentence is revised:

A type k thermocouple, which is blue electrical wire in Figure 7, was wound on the fiber surface to check the temperature variation

  1. page 8: conjectured -à deemed

  1. page 8: The below paragraph is revised:

It is deemed that the fibers did not undergo complete phase transformation because the surface fiber temperature of 100℃ around the type k thermocouple was slightly higher than the A(91.46℃), where the phase transformation to austenite is completed. Therefore, it seems that all parts of the fiber may not have undergone phase transformation.

-à It is deemed that the fibers did not undergo complete phase transformation because the heating was stopped just after the type k thermocouple reached 100℃, and thus all parts of the fiber seemed not to arrive at 100℃. Therefore, it seems that all parts of the fiber may not have undergone phase transformation. This means that heating temperature higher than 100℃ can induce more phase transformation and thus increase the recovery stress.

  1. page 6: For the prestressing effect,~~~ à the sentence is revised:

although, for the prestressing effect, the fiber could be heated with a higher temperature to have more benefits.

  1. page 8: The maximum recovery and residual stresses of the CR0.7 fibers also showed an increasing trend with increasing temperature; however, these values were disturbed.--> the sentence is revised:

Response: The maximum recovery and residual stresses of the CR0.7 fibers also showed an increasing trend with increasing temperature. However, the maximum recovery and residual stresses of the CR0.7 fibers are not proportional to the wave height as like the CR1.0 fibers do. It seems that manufacturing of the CR0.7 fiber did not conduct exactly to control the wave height because the gap between the two gears is too small to precisely control. Thus, it is may possible for the CR2-0.7 fiber to show the largest recovery and residual stress among the CR0.7 fibers.

  1. page 10:

Comment: recommend different line types for each temperature plot; - Correct: (j) CR0.7-4; - Use a consistent font size for figure legends, axis tick values

Response: The figure is revised.

Figure 8. Temperature-stress curves of AR and crimped fibers

  1. page 11:

Comment: "time-temperature history"

- is the temperature measured using type k thermocouple or the temperature history of heat chamber?

Response:

A type k thermocouple (blue electrical wire in Figure 10a) was wound on the surface of the fiber embedded in the mortar matrix to measure the temperature. Figure 10b shows time-temperature history of the heated specimen.

  1. page 11: The temperature was increased rapidly in the beginning, and the increment became blunt after one hour and became stable at 145℃ in three hours.

Response: The sentence is revised:

The temperature of the fiber buried in mortar increased rapidly for an hour. After that, the increment of temperature became blunt for two hours and stable with 145℃ after three hours.

  1. page 11: the fiber was kept with a 5 mm length from the top surface of the mortar specimen by an actuator.

Response: the fiber was hold by the wedge grips at 5 mm length from the top surface of the specimen.

  1. page 16: in the duct à The related paragraph is revised like below:

Response:

This phenomenon is highly related to flexural stiffness of the bent part in a crimped fiber. The flexural stiffness of the bent part is function of wave depth, which acts like a cantilever arm in bending, and flexural rigidity (EI) of the fiber. Thus, if both types of CR0.7-N and CR1.0-N fibers have the same wave depth, the flexural stiffness of the CR1.0-N fiber have four times larger than that of the CR0.7-N fiber. Therefore, it can be said for the crimped SMA fiber that the maximum pullout force is highly related to the flexural stiffness of the bent part.

  1. page 18: Crimped fibers show lower recovery stress and residual stress~~ à The sentence is revised:

Response:

  • Crimped fibers by heating show lower recovery stress and residual stress in comparison with AR fiber having the same initial diameter because of the stretching effect. For CR1.0, the fibers with lower wave height showed higher values of recovery and residual stresses; however, the stresses of CR0.7 fibers were disturbed with increasing temperature.

  1. page 18: The compressive strength of the mortar--> The compressive strength of cement matrix

Response: The sentence is general comment. Thus, the mortar is changed to cement matrix to avoid confusing.